# Biases in estimated vegetation indices from observations under cloudy conditions

Kevin Wolf[1], Evelyn Jäkel[1], André Ehrlich[1], Michael Schäfer[1], Hannes Feilhauer[2,3,4], Andreas Huth[4,5,6], and Manfred Wendisch[1]

[1]Leipzig Institute for Meteorology (LIM), Leipzig University, Leipzig, Germany
[2]Institute for Earth System Science & Remote Sensing, Leipzig University, Leipzig, Germany.
[3]Remote Sensing Centre for Earth System Research, Leipzig University, Leipzig, Germany.
[4]iDiv German Centre for Integrative Biodiversity Research Halle-Jena-Leipzig, Leipzig, Germany.
[5]Department of Ecological Modelling, Helmholtz Centre for Environmental Research–UFZ Leipzig, Leipzig, Germany.
[6]Institute for Environmental Systems Research, University of Osnabrück, Osnabrück, Germany.

**Correspondence:** Kevin Wolf (kevin.wolf@uni-leipzig.de)

**Abstract.** Field observations of vegetation indices (VIs) are derived from ratios of spectral reflectance data that are collected by drones and aircraft, providing higher spatial resolution than satellites. These reflectance data require periodic reference measurements over calibrated reflectance panels under cloud-free conditions. However, the reference measurements are partly performed in cloudy situations with the effect that wavelength-dependent scattering and absorption of solar radiation by clouds affects the subsequently derived VIs. This paper quantifies these effects using combined atmosphere-vegetation radiative transfer (RT) simulations. We study the general case when VIs are obtained from reflectance ratios of two wavelengths, and for the special cases of the normalized difference vegetation index (NDVI), the normalized difference water index (NDWI), and the enhanced vegetation index (EVI). For the general case of two-band VIs the lowest sensitivity to cloud changes was found for wavelength combinations below 1400 nm and outside the water vapor absorption bands. The NDVI was almost insensitive to changes in cloud conditions, while greater biases were identified for the NDWI. The EVI was most susceptible to cloud changes, with biases of 0.2 in the selected example. This lead to biases in the estimated leaf area index of 0.9. Biophysical properties derived from EVI, such as gross primary product, are also affected with variations of up to $\pm 2\ \mathrm{gCm^{-2}\,d^{-1}}$ in the selected cases.

## 1 Introduction

Numerous vegetation indices (VIs) have been developed to remotely characterize vegetation. They are based on ratios of vegetation-reflected radiation in wavelength regions that are sensitive to chlorophyll and liquid water absorption, thus indirectly providing information on plant vitality, productivity, and photosynthetic activity, and to monitor anthropogenic degradation (Collins, 1978; Horler et al., 1983; Bowker, 1985; Myneni et al., 1997; Saleska et al., 2007; Jiang et al., 2008; Knyazikhin et al., 2013; Xue and Su, 2017; Mura et al., 2018; Richardson et al., 2021). Established VIs are, for example, the normalized difference vegetation index (NDVI), the normalized difference water index (NDWI), and the enhanced vegetation index (EVI) (Kriegler et al., 1969; Huete et al., 1994; Myneni et al., 1995; Gao, 1996; Chen et al., 2005; Jiang et al., 2008; Jones and

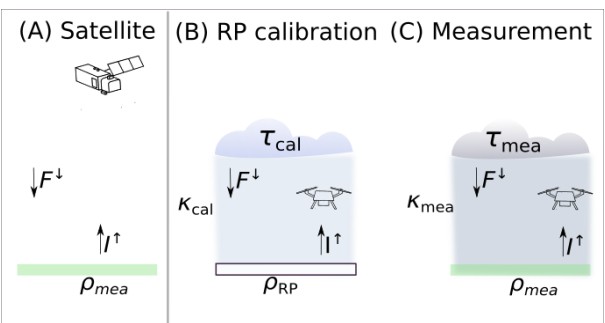

**Figure 1. (A)** Schematic illustration of measured spectral upward radiance $I^\uparrow$ under cloud-free conditions. **(B)** and **(C)** Schematic illustration of below cloud measurements that can be subject to a change in cloud conditions from $\tau_{\text{cal}}$ (during reflectance panel (RP) calibration) to cloud conditions $\tau_{\text{mea}}$ (during actual measurements). The spectral total downward irradiance $F^\downarrow$ and spectral upward radiance $I^\uparrow$ are affected by the cloud conditions. The reflectance $\rho_{\text{RP}}$ of the RP panel is known from its specifications and $\rho_{\text{mea}}$ is the reflectance of the vegetation. The transfer functions $\kappa_{\text{cal}}$ and $\kappa_{\text{mea}}$ connect measured counts and $\rho$.

Vaughan, 2010). They were originally developed for satellite observations and customized to spectral bands of satellites such as Landsat (Wulder et al., 2019), the Moderate Resolution Imaging Spectroradiometer (MODIS; Salomonson et al., 1989), or Sentinel-2 (Spoto et al., 2012). In the course of the time, the available measurement platforms diversified, additionally 25 including aircraft and unmanned aerial vehicles (UAVs, Matese et al., 2015; Duan et al., 2017; Singh and Frazier, 2018; Jiang et al., 2020).

All remotely sensed VIs rely on the spectral reflectance $\rho(\lambda)$. In its most general form, $\rho(\lambda)$ is defined as the ratio of the surface-reflected radiance from within an infinite solid angle to the incoming radiance from within an infinite solid angle (Nicodemus, 1977; Martonchik et al., 2000; Schaepman-Strub et al., 2006). The amount of radiation reflected into a given 30 solid angle is defined by the surface-specific bidirectional reflectance distribution function (BRDF, Schaepman-Strub et al., 2006). Furthermore, the surface reflected radiation depends on topography (Matsushita et al., 2007) and on changes in illumination conditions determined by solar zenith angle, aerosol particles, and clouds (Pinty et al., 2005; Singh and Frazier, 2018). Definitions of $\rho(\lambda)$ based on first principals are given by Nicodemus (1977) and Schaepman-Strub et al. (2006). For brevity and simplicity, we restrict the definition of $\rho(\lambda)$, assuming idealized conditions of pure diffuse illumination and Lambertian 35 reflection at the surface, which results in:

$$\rho(\lambda) = \frac{I^\uparrow(\lambda) \cdot \pi \cdot \text{sr}}{F^\downarrow(\theta, \lambda)}, \tag{1}$$

where $\lambda$ symbolizes the wavelength, sr the unit steradian, $I^\uparrow(\lambda)$ the spectral, upward (reflected) radiance in units of $\text{W m}^{-2}\,\text{nm}^{-1}\,\text{sr}^{-1}$, and $F^\downarrow(\theta, \lambda)$ the downward spectral irradiance in units of $\text{W m}^{-2}\,\text{nm}^{-1}$, which also depends on the solar zenith angle $\theta$. The cloud optical thickness $\tau(\lambda)$ is a measure of the extinction of radiation for a vertical path through the cloud, serving as ver- 40 tical coordinate. Radiation that has never been scattered represents the direct radiation component $F_{\text{dir}}^\downarrow(\theta, \lambda)$. The spectral reflectance ranges between 0 and 1, except under broken cloud conditions where radiation enhancement occurs, leading to

reflectance values exceeding unity (Schaepman-Strub et al., 2006; Marshak et al., 2008). Only in heterogeneous cloud cases, where three-dimensional scattering effects occur, are reflectance values greater than one observable. In neighboring regions, reflectance is reduced. On average, this results in energy conservation. Radiation that has undergone at least one scattering event contributes to diffuse irradiance $F_{\mathrm{dif}}^{\downarrow}(\theta, \tau, \lambda)$. Thus, $F^{\downarrow}(\theta, \tau, \lambda)$ is decomposed into:

$$F^{\downarrow}(\theta, \tau, \lambda) = F_{\mathrm{dir}}^{\downarrow}(\theta, \tau, \lambda) + F_{\mathrm{dif}}^{\downarrow}(\theta, \tau, \lambda). \tag{2}$$

The related direct fraction $f_{\mathrm{dir}}(\theta, \tau, \lambda)$ is defined by:

$$f_{\mathrm{dir}}(\theta, \tau, \lambda) = \frac{F_{\mathrm{dir}}^{\downarrow}(\theta, \tau, \lambda)}{F^{\downarrow}(\theta, \tau, \lambda)}. \tag{3}$$

Remote sensing techniques to estimate VIs from satellite measurements require cloud-free conditions. Cloud masks are used to flag cloud-contaminated pixels, which are excluded from the VI calculation. The remaining pixels are assumed to be unbiased by clouds. Figure 1 (A) schematically illustrates the measurement principle. However, for all types of airborne measurements from drones or aircraft that are maybe performed below clouds, see Fig. 1 (B and C), the downward spectral irradiance $F^{\downarrow}(\lambda)$ is altered by the presences of clouds. The impact of clouds depends on $\tau$, droplet/ice crystal size, and wavelength (Twomey and Cocks, 1989). The contribution of scattering and absorption is spectrally dependent, with scattering most pronounced in the visible near–infrared (VNIR, 0.3–1.0 $\mu$m) part of the solar spectrum (Mie, 1908; van de Hulst, 1981; Dubovik et al., 2002) and absorption in the shortwave infrared (SWIR, 1.0–2.5 $\mu$m) wavelength range. The spectral dependency leads to a spectral slope in $F^{\downarrow}(\lambda)$, $I^{\uparrow}(\lambda)$, and $\rho(\lambda)$ (Wiscombe and Warren, 1980; Grenfell and Perovich, 2008). Consequently, changes in cloud properties and illumination lead to biases in estimated VIs.

To obtain $\rho(\lambda)$ for VI retrievals during field studies, calibrated $I^{\uparrow}(\tau, \lambda)$ and $F^{\downarrow}(\theta, \tau, \lambda)$ are required, which are sometimes measured with dedicated, calibrated sensors (Hakala et al., 2013; Honkavaara et al., 2013; Miyoshi et al., 2018). Alternatively, $F^{\downarrow}(\theta, \tau, \lambda)$ is simulated using radiative transfer (RT) models. In either case, absolute measurements are complicated and simulations introduce uncertainties. For practical applications, absolute measurements are often avoided by performing relative measurements, using reflectance panels (RPs) characterized by a well-known reflectance $\rho_{\mathrm{RP}}(\lambda)$ (Burkart et al., 2014; Aasen et al., 2015; Aasen and Bolten, 2018; Hakala et al., 2018; Fawcett et al., 2020; Köppl et al., 2021).

Using a RP allows a transfer calibration. Digital spectrometers register digital counts $S$, which are related to $I^{\uparrow}(\tau, \lambda)$ by a calibration factor $\mathcal{C}(\lambda)$. Including this calibration factor in Eq. 1 yields:

$$\rho(\lambda) = \frac{S^{\uparrow}(\tau, \lambda) \cdot \mathcal{C}^{\uparrow}(\lambda) \cdot \pi \cdot \mathrm{sr}}{F^{\downarrow}(\theta, \tau, \lambda)}, \tag{4}$$

where $S^{\uparrow}(\tau, \lambda)$, $I^{\uparrow}(\tau, \lambda)$, and $F^{\downarrow}(\theta, \tau, \lambda)$ below clouds dependent on $\tau$. Equation 4 can be simplified to:

$$\rho(\lambda) = S^{\uparrow}(\tau, \lambda) \cdot \kappa(\theta, \tau, \lambda), \tag{5}$$

with the transfer function $\kappa(\theta, \tau, \lambda)$ given by:

$$\kappa(\lambda, \tau, \theta) = \frac{\mathcal{C}^{\uparrow}(\lambda) \cdot \pi \cdot \mathrm{sr}}{F^{\downarrow}(\tau, \lambda)}. \tag{6}$$

In a first step during the relative measurements, the RP is overflown and the signal is taken for calibration (Fig. 1, B), where $\rho(\lambda)$ in Eq. 5 is set to the well-defined $\rho_{\text{RP}}(\lambda)$ and $S^{\uparrow}(\lambda)$ is the number of counts registered by the radiance sensor above the RP. The transfer function $\kappa_{\text{cal}}(\theta, \tau_{\text{cal}}, \lambda)$ determined during the calibration procedure now represents the relationship between the recorded digital counts and the surface reflectance under the illumination conditions at that time. During actual measurements over vegetation (Fig. 1, C), the reflectance $\rho_{\text{mea}}(\lambda)$ of the vegetated surface is determined applying $\kappa(\theta, \tau_{\text{cal}}, \lambda)$:

$$\rho_{\text{mea}}(\lambda) = \kappa_{\text{cal}}(\theta, \tau_{\text{cal}}, \lambda) \cdot S^{\uparrow}(\lambda). \tag{7}$$

Although RP calibration avoids the determination of the individual factors included in $\kappa_{\text{cal}}(\theta, \tau_{\text{cal}}, \lambda)$, the dependence of $\kappa(\theta, \tau, \lambda)$ on $\theta$ and $\tau$, requires frequent RP overflights to account for illumination changes and to obtain updated transfer functions.

Lohmann (2018) presented example time series of $F^{\downarrow}$ from a field campaign (Madhavan et al., 2016; Macke et al., 2017) and compared them with cloud-free simulations. They determined single-point enhancements of $F^{\downarrow}$ up to $50\,\%$ due to cloud-side enhancements. Short-term fluctuations of $F^{\downarrow}$ have been found on time scales down to $300\,$s (Jurado et al., 1995), $60\,$s (van Haaren et al., 2014), $20\,$s (Perez et al., 2011), and also below $1\,$s (Calif et al., 2013). Consequently, $\kappa(\theta, \tau, \lambda)$ and thus $\rho_{\text{mea}}(\lambda)$ vary on the same time scales. Therefore, even regular sequences of RP measurements seem insufficient to capture fluctuations caused by broken clouds, either by the clouds themselves or by their shadows, which may lead to potential errors in the retrieved VIs (Burkart et al., 2014; Behmann et al., 2015; Köppl et al., 2021). Even when $F^{\downarrow}(\lambda)$ is measured by a dedicated sensor or frequent measurements of the RP are performed, the sole presence of clouds causes a change in $\rho_{\text{mea}}(\lambda)$, since clouds change the ratio of direct and diffuse radiation, which determines how radiation is reflected by a non-isotropic surface (Schaepman-Strub et al., 2006).

Here we present coupled simulations using the atmospheric RT model "library for Radiative transfer" (libRadtran Emde et al., 2016) and the vegetation RT model "Soil Canopy Observation of Photosynthesis and Energy fluxes" (SCOPE2.0 Yang et al., 2021; Wolf et al., 2025a). The iterative model coupling allows to account for mutual influences between cloud and canopy. The coupled simulations are used to systematically investigate the impact of different cloud conditions on general two-band VIs, and in particular on NDVI, NDWI, and EVI. The aim is to quantify the impact of clouds on below-cloud airborne observations that are based on relative measurements using RP calibration thus providing approximate associated errors in estimated VI. The VIs are calculated using the wavelength bands of the multispectral instrument (MSI) onboard the Sentinel-2 satellite. This introduction is followed by a brief overview of the main terminology, the applied RT models, and a short review of the model coupling in Section 2. This is followed by the results in Section 3, where the effect of clouds and changing cloud conditions on estimated VIs are presented. Section 4 provides a summary of the results and outlines the limitations of the simulations.

**Table 1.** Sentinel-2 wavelength bands from the multispectral instrument (MSI) following Drusch et al. (2012).

| Band number | Center wavelength [nm] | Spectral width [nm] |
|:-----------:|:----------------------:|:-------------------:|
| B2  | 490  | $\pm 33$  |
| B3  | 560  | $\pm 35$  |
| B4  | 665  | $\pm 15$  |
| B8  | 842  | $\pm 56$  |
| B8a | 865  | $\pm 10$  |
| B9  | 945  | $\pm 10$  |
| B10 | 1375 | $\pm 15$  |
| B11 | 1610 | $\pm 45$  |
| B12 | 2190 | $\pm 180$ |

## 2 Vegetation indices and radiative transfer simulations

### 2.1 Definition of vegetation indices

Vegetation indices are based on ratios of $\rho(\lambda)$ at several wavelengths (mostly pairs). The exact center wavelength and width used in the calculation of VIs depend on the characteristics of the observing instrument. In the case of the Sentinel-2 satellites, several wavelength combinations are suitable for VI retrievals (Kriegler et al., 1969; Carlson and Ripley, 1997; Mura et al., 2018). A subset of the Sentinel-2 wavelength bands are listed in Table 1. Unless otherwise noted, the Sentinel-2 wavelength bands were used to calculate VIs throughout this paper by applying a boxcar-like spectral response function. The Sentinel-2 wavelength bands are considered representative of other measurement platforms and sensors. UAVs may be equipped with spectral sensors that cover similar wavelength bands, or in the case of multispectral sensors, spectral integration is performed over similar wavelength bands (Jones et al., 2012).

Multiple two-band VIs with the index value $\gamma$ exist that follow the general form of a band transformation:

$$\gamma = \frac{\rho(\lambda_1) - \rho(\lambda_2)}{\rho(\lambda_1) + \rho(\lambda_2)}, \tag{8}$$

where $\lambda_1$ and $\lambda_2$ represent a pair of individual wavelengths or narrow bands with band centers between 400 and 2400 nm wavelengths.

An example of a two-band VI is the normalized difference water index (NDWI). Two versions of the NDWI exist. The NDWI proposed by Gao (1996) uses a wavelength combination of 980 and 1240 nm; it is subsequently labeled with $\mathrm{NDWI}_{1240}$. Chen et al. (2005) provided the alternative $\mathrm{NDWI}_{1640}$, which was designed to be less sensitive to saturation with respect to the water content in plant matter compared to $\mathrm{NDWI}_{1240}$. Another commonly used VI that follows Eq. 8 is the normalized difference vegetation index (NDVI; Kriegler et al., 1969), with $\rho(\lambda_1)$ from B4 and $\rho(\lambda_2)$ from B8 from the MSI Sentinel-2 channels. By construction, the NDVI ranges between $-1$ and $1$. An NDVI close to unity indicates vital and productive vegetation,

while stressed vegetation results in NDVI values around 0.2. Measurements above bare soil will give a NDVI around 0, while measurements over water will yield negative NDVI. Even though the NDVI proofed to be useful it also showed sensitivity to surface brightness, and scattering and absorption by atmospheric constituents (Huete et al., 1985; Kaufman and Tanre, 1992; Bausch, 1993; Miura et al., 1998). The NDVI also becomes saturated and insensitive to vegetation structure when the canopies become dense, i.e., high leaf density (Ünsalan and Boyer, 2011).

These issues led to the development of the enhanced vegetation index (EVI; Huete et al., 1994), which was derived to overcome the shortcomings of the NDVI (Boegh et al., 2002; Huete et al., 2006). The EVI attempts to account for aerosol loading, surface reflectance, and other factors by including spectral information from the "blue" region ($\rho_{B2}$) of the VNIR spectrum. The EVI is calculated for MSI Sentinel-2 bands by:

$$\text{EVI} = G \cdot \frac{\rho_{B8} - \rho_{B4}}{\rho_{B8} + C_1 \cdot \rho_{B4} - C_2 \cdot \rho_{B2} + L}, \tag{9}$$

with the scaling factors $G = 2.5$, $C_1 = 6$, $C_2 = 7.5$, and $L = 1$ (Liu and Huete, 1995).

## 2.2 Radiative transfer simulations

Equation 1 shows that the reflectance of a surface is determined by several factors, including $\theta$, the observation geometry, as well as the direct and diffuse components of $F^\downarrow(\lambda)$ that are determined by $\tau$ (see Section B in the appendix). Furthermore, radiation interactions may occur between the surface and the cloud, which can be accounted for by iterative coupling of the RT models of the atmosphere and vegetation (Wolf et al., 2025a). In the present paper we use the same model coupling setup introduced and described by Wolf et al. (2025a).

### 2.2.1 Atmospheric radiative transfer model libRadtran

The atmospheric RT above the canopy was simulated with the library for Radiative transfer (libRadtran, Emde et al., 2016). The RT equation was calculated with the one-dimensional solver "Discrete-Ordinate-Method Radiative Transfer" (DISORT, Stamnes et al., 1988; Buras et al., 2011) using 12 streams. Clouds were assumed to be homogeneous. A low-level and mid-level warm stratus or altostratus were included by liquid water clouds between 3 and 3.5 km altitude with a fixed droplet effective radius of 10 µm (Stephens, 1994; Frisch et al., 2002; Aebi et al., 2020). A high-level cirrostratus was included by ice water clouds between 11 and 11.5 km altitude with a fixed ice effective radius of 85 µm (Freudenthaler et al., 1995; Sassen and Campbell, 2001; Noël and Haeffelin, 2007; Iwabuchi et al., 2012; Luebke et al., 2016; Krämer et al., 2016). Aggregated ice particles with moderate surface roughness are assumed to represent mature ice crystals (Liu et al., 2014; Holz et al., 2016; Järvinen et al., 2018). The ice particle parametrization after Yang et al. (2013) was applied. The liquid water and ice water path were scaled so that a specific $\tau(\lambda = 550\,\text{nm})$ at 550 nm wavelength was reached, with all other wavelengths being scaled considering the wavelength dependence of $\tau$. For simplicity, $\tau(\lambda = 550\,\text{nm})$ is referred to as $\tau$ in the following. The incoming spectral irradiance at the top of atmosphere was represented by the solar reference spectrum provided by Coddington et al. (2021). Molecular absorption was considered using the parameterization of the "medium" resolution of Gasteiger et al. (2014). A default aerosol distribution after Shettle (1989) was applied, representing rural type aerosol in the boundary layer, back-

ground aerosol above 2 km during spring-summer, and a visibility of 50 km. Atmospheric profiles of air temperature, humidity, and gas concentrations were taken from the mid-latitude summer profile 'afglms' (Anderson et al., 1986). Absorption by water vapor and other atmospheric trace gases was accounted for in the simulations (Anderson et al., 1986; Emde et al., 2016). The iteration process was first started by running libRadtran with an initial guess for the surface albedo. The "mixed-forest" albedo was taken from the IGBP database (Loveland and Belward, 1997). After one iteration cycle, the surface albedo determined during the iterative model coupling process was used (Wolf et al., 2025a). The RT simulations of $F^\uparrow(\lambda)$, $F^\downarrow(\lambda)$, and $I^\uparrow(\lambda)$ spanned a wavelength range from 0.4 to 2.4 μm. The output of libRadtran was specified to be at an altitude of 40 m above the canopy to be representative for UAV measurements. The radiances were simulated for a nadir sensor geometry.

### 2.2.2 Vegetation radiative transfer model SCOPE2.0

The solar RT within vegetation was simulated with the Soil Canopy Observation of Photosynthesis and Energy fluxes version 2 (SCOPE2.0, Yang et al., 2017, 2020, 2021). The surface albedo was determined using the Brightness-Shape-Moisture (BSM) model (Verhoef et al., 2018; Yang et al., 2020). Irradiance and radiance simulations in the solar part of the spectrum have been performed for a wavelength range from 0.4 to 2.4 μm. The angular dependence of $I^\uparrow(\lambda)$ is considered for by the actual illumination and observation geometries, the direct and diffuse $F^\downarrow(\lambda)$, and the parameterized intrinsic reflectivity within SCOPE2.0. An important parameter to describe the radiative properties of a canopy is the leaf area index (LAI, Watson, 1947; Asner, 1998; Jones and Vaughan, 2010), typically ranging between 0 and 12. It provides a measure of the accumulated, one-sided area of leaves per unit of ground area given in units of $m^2\,m^{-2}$. A constant LAI of 3 $m^2\,m^{-2}$ was used in the simulations, corresponding to the typical LAI of temperate vegetation types such as pine forests or various grasslands. The leaf angle distribution (LAD) is another important parameter controlling the RT in a canopy (Baldocchi et al., 2002; Jones and Vaughan, 2010; Verrelst et al., 2015; Yang et al., 2023). It is a measure of the overall orientation of the leaf ensemble of a tree. Therefore, it influences the area of a leaf illuminated by the Sun with respect to the total one-sided leaf area (Asner, 1998; Stuckens et al., 2009; Vicari et al., 2019). The simulations were primarily considered for the general case of a spherical LAD, where all leaf angles have a similar probability (Goel, 1988). Further simulations have been performed to also include the erectophile and the planophile LADs. The static LAD values obviously ignores that short term changes in weather and illumination conditions affect the LAD (Kattenborn et al., 2022), which may likewise have significant effects on VIs (Kattenborn et al., 2024), but is nevertheless common practice in many RTM parameterizations. Table 2 provides an overview of the selected parameters for the vegetation RT simulations. The parameters were selected on the basis of their relevance to surface reflectance within the visible and near-infrared wavelength ranges. Their individual relevance was estimated in a sensitivity study by Wolf et al. (2025a).

### 3 Results and discussion

The one-dimensional simulations combining libRadtran and SCOPE2.0 are to be interpreted as synthetic measurements of $F^\downarrow(\lambda)$, $I^\uparrow(\lambda)$, and $\rho(\lambda)$ under varying atmospheric conditions, ranging from cloud-free to stratiform clouds, with values of

**Table 2.** Selected configuration of the SCOPE2.0 simulations.

| Description | Symbol | Setting | Unit |
|---|---|---|---|
| Leaf chlorophyll concentration | $C_{\mathrm{ab}}$ | 40 | µg cm$^{-2}$ |
| Leaf carotenoid concentration | $C_{\mathrm{ca}}$ | 10 | µg cm$^{-2}$ |
| Leaf water equivalent layer | $C_{\mathrm{w}}$ | 0.009 | cm |
| Leaf structure parameter | $N$ | 2.1 | Unitless |
| BSM model parameter for soil brightness | $B$ | 0.5 | Unitless |
| Vegetation height | $h_{\mathrm{c}}$ | 20 | m |
| Output height | $h_{\mathrm{out}}$ | 40 | m |

$\tau$ between 0 and 40, and $\theta$ between 25° and 70°. By assuming various combinations of $\tau_{\mathrm{cal}}$, which prevailed during calibration and $\tau_{\mathrm{mea}}$, present during actual measurements, the effects of changes in cloud conditions between RP measurements on $\rho_{\mathrm{mea}}(\lambda)$ and estimated VIs were determined. This was performed using simulated $I^{\uparrow}(\tau_{\mathrm{mea}}, \lambda)$ and $F^{\downarrow}(\tau_{\mathrm{cal}}, \lambda)$ in Eq. 1, which is synonymous with the assumption of constant values of $\kappa_{\mathrm{cal}}$. The values of simulated $\tau_{\mathrm{cal}}$ and $\tau_{\mathrm{mea}}$ range from 0 to 40 for the liquid water cloud and the ice water cloud. Two aspects influenced the chosen ranges. First, simulated ice clouds with $\tau$ up to 6 can be considered as a high-level cirrus. The natural variation in non-spherical ice particle shapes was taken into account by selecting aggregated ice particles (see Sec.2.2.1). Second, liquid water clouds were considered by simulating mid-level, continental stratus clouds, which are known to often reach values of up to $\tau = 40$ (King et al., 1993; Lu et al., 2008). The subsequent analysis focuses primarily on the spherical LAD, which is considered the general case. The effect of the selected LAD is discussed in Appendix Sec. A.

## 3.1 Influence of clouds on the spectral reflectance

Pinty et al. (2005) have shown the influence of molecules and aerosols on direct and diffuse $F^{\downarrow}(\lambda)$, the effects on $F^{\uparrow}(\lambda)$, and the resulting albedo effects over vegetated areas based on analytical equations. Wolf et al. (2025a) used coupled atmosphere–vegetation radiative transfer models to investigate these effects, explicitly considering clouds and spectral depending scattering and absorption in the atmosphere. An increase in $\tau$ leads to a decrease in $F^{\downarrow}_{\mathrm{dir}}(\lambda)$, while the response of $F^{\downarrow}_{\mathrm{dif}}(\lambda)$ depends on $\tau$. For values of $\tau$ less than 4 to 6, $F^{\downarrow}_{\mathrm{dif}}(\lambda)$ first increases and then decreases as $\tau$ increases further. The total $F^{\downarrow}(\lambda)$ and $f_{\mathrm{dir}}(\lambda)$ both continuously decrease as $\tau$ increases. In addition, $F^{\uparrow}(\lambda)$ became less sensitive to changes in $\theta$. Lastly, the presence of clouds modulates the incoming radiation spectrally by shifting the incoming radiation towards shorter wavelengths, as clouds primarily scatter radiation at shorter wavelengths and absorb radiation at longer wavelengths. Wolf et al. (2025a) also showed that radiative interactions between the canopy and the cloud base increase $F^{\downarrow}_{\mathrm{dif}}(\lambda)$ and albedo compared to cloud-free conditions. The present paper focuses on the related effects on $I^{\uparrow}(\lambda)$ and $\rho(\lambda)$.

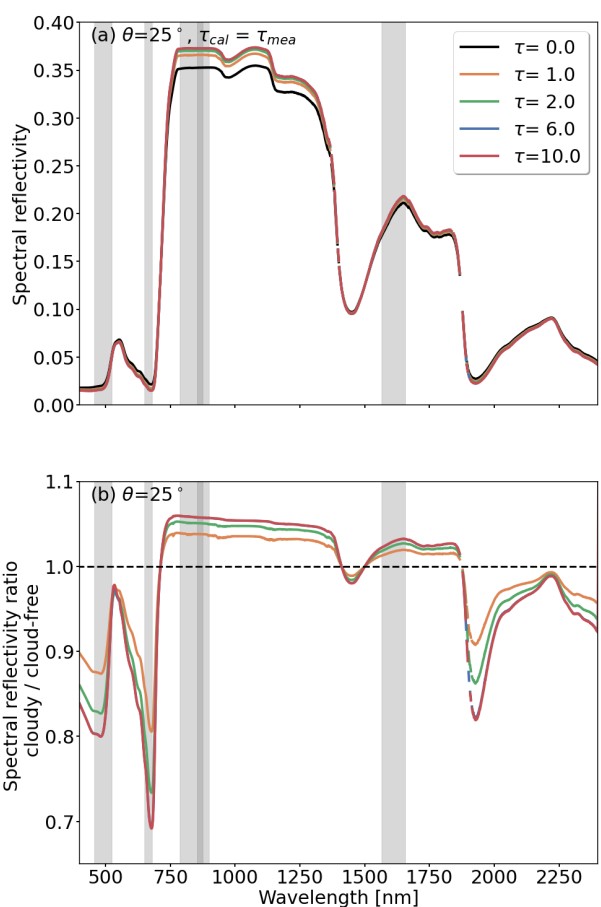

**Figure 2.** (a) Simulated spectral reflectance $\rho(\lambda)$ caused by an ice cloud for a solar zenith angle $\theta$ of 25°, and constant cloud optical thickness during calibration and the actual synthetic measurements ($\tau_{\text{cal}} = \tau_{\text{mea}}$). The simulations base on a spherical LAD. (b) Ratios of spectral reflectance with $\tau_{\text{cal}} = \tau_{\text{cal}} > 0$ with respect to the spectral reflectance under cloud-free conditions. Panels (a) and (b) share the same legend. The gray marked areas highlight the Sentinel-2 bands B2, B4, B8, B8a, and B11.

### 3.1.1 Effect of diffuse radiation on spectral reflectance above vegetation

First, we consider situations, where the cloud conditions are similar during the RP calibration and the measurements ($\tau_{cal} = \tau_{mea}$). Figure 2a shows $\rho(\lambda)$ for $\theta = 25°$ with the lowest $\rho(\lambda)$ under cloud-free conditions ($\tau_{cal} = \tau_{mea} = 0$, black line). With increasing values of $\tau$, $\rho(\lambda)$ also increases, especially between 750 and 1300 nm wavelengths, where vegetation is generally characterized by high reflectivity and reflectance. The sensitivity of $\rho(\lambda)$ to $\tau$ is highest for small values of $\tau$ and quickly approaches an asymptotic value when $F^\downarrow_{dif}(\lambda)$ dominates the radiation field. Figure 2b shows the ratio of $\rho(\lambda)$ between cloudy and cloud-free conditions. Independent of $\tau$, the ratio has a distinct spectral dependency with pronounced water absorption bands. Under cloudy conditions ($\tau = 10$) and for $\theta = 25°$, $\rho(\lambda)$ decreases by up to 30 % for wavelength below 750 nm, while for longer wavelengths $\rho(\lambda)$ increases by up to 8 % at about 780 nm. An exception is the dip in $\rho(\lambda)$ at about 1450 nm wavelength and wavelengths greater than 1800 nm. For $\theta > 60°$, an increase in $\tau$ leads to an increase in $\rho(\lambda)$ throughout the simulated wavelength range.

The response of $\rho(\lambda)$ with increasing $\tau$ results from the transition from only direct to only diffuse radiation, which is controlled by the combination of $\theta$ and $\tau$. Two theoretical extremes are distinguished: i) only direct radiation, also called "black-sky", and ii) only diffuse radiation, also called "white-sky". Natural conditions typically lie between these two extremes; they are referred to as "blue-sky" (Lucht et al., 2000). The amount of direct radiation controls two effects. The first effect acts on the canopy level, where diffuse radiation can penetrate deeper into the canopy and interacts with leaves that would be shaded under black-sky conditions. Consequently, for the same LAD and LAI, a greater total leaf area interacts with the incoming radiation (Jarvis et al., 1985; Freedman et al., 2001). Model simulations by Wolf et al. (2025a) showed an increase in broadband solar canopy albedo with increasing $f_{dir}$. Furthermore, the extinction of radiation is sensitive to the incident angle, which is equal to $\theta$ for direct radiation and approaches an effective value of about 60° under overcast conditions, due to the increasing contribution from diffuse radiation (Pinty et al., 2005; Gardner and Sharp, 2010). The second factor acts on the individual leaf level, where the inherent directional reflection of radiation on surfaces is relevant. In the case of non-isotropic surfaces, direct radiation is scattered in all directions with a significant portion of the radiation being scattered in a specific solid angle. Diffuse radiation is scattered almost equally over the entire hemisphere. (Asner, 1998; Schaepman-Strub et al., 2006). Both effects are directly linked to the directional reflectance under blue-sky conditions that is described by the hemispherical–directional reflectance factor (HDRF, Schaepman-Strub et al., 2006), which includes the influence of the diffuse component in $F^\downarrow(\lambda)$. The reflection on the leaf-level combined with the LAD then determines the HDRF of the entire canopy. An example of a HDRF for a spherical LAD is given in Appendix B.

### 3.1.2 Effect of cloud changes on spectral reflectance above vegetation

The effects presented above, do not include changes in cloud conditions between RP calibration and actual measurement over vegetated surfaces, where $\tau_{cal} \neq \tau_{mea}$ and $F^\downarrow_{cal}(\lambda) \neq F^\downarrow_{mea}(\lambda)$. The effects of cloud changes between RP measurements to $F^\downarrow_{mea}(\lambda)$ are quantified and shown in Fig. 3, providing the illumination ratio $F^\downarrow_{mea}(\lambda)/F^\downarrow_{cal}(\lambda)$. The illumination ratio is calculated between the irradiance $F^\downarrow_{cal}(\lambda)$ that prevailed during the RP calibration and the irradiance $F^\downarrow_{mea}(\lambda)$ that is present

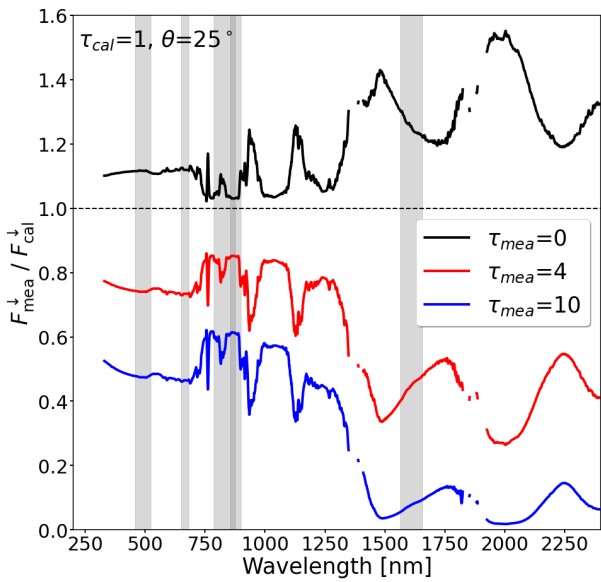

**Figure 3.** Ratio of downward irradiance $F_{\mathrm{mea}}^{\downarrow}(\lambda)$ present during the synthetic measurements and downward irradiance $F_{\mathrm{cal}}^{\downarrow}(\lambda)$ present during the calibration. The ratio is calculated for ice clouds, where $\tau_{\mathrm{cal}} = 1$ and $\tau_{\mathrm{mea}}$ is set to 0, 4, and 10. As an example, a solar zenith angle $\theta$ of $25°$ is selected. The gray marked areas highlight the Sentinel-2 bands B2, B4, B8, B8a, and B11.

during the above-vegetation measurements. The given illumination ratio represents the conditions below an ice cloud with $\tau_{\mathrm{cal}} = 1$ and three values of $\tau_{\mathrm{mea}}$ of 0, 4, and 10. In general, measurements where $\tau_{\mathrm{cal}} < \tau_{\mathrm{mea}}$ yield an illumination ratio less than one, because the reflectance at cloud top and the absorption inside the cloud become more intense, reducing the total $F^{\downarrow}(\lambda)$ below the cloud. For opposite conditions, where $\tau_{\mathrm{cal}} > \tau_{\mathrm{mea}}$, the illumination ratio is greater than one, since scattering and absorption are less during actual measurement compared to the RP calibration. Clearly visible are the ice absorption features,

for example at 1500 nm wavelength, and a generally larger sensitivity of the illumination ratio towards longer wavelengths. Similar illumination ratios are calculated for liquid water clouds but with lower sensitivity for same values of $\tau$. This is due to the smaller cloud droplet size of 10 μm compared to the ice particle size used in the simulations and the difference in the single-scattering phase function between liquid water droplets and ice crystals. In addition, differences in the imaginary part of the refractive indices lead to a slight spectral shift in the absorption features (Pilewskie and Twomey, 1987).

The spectral distortion in $F^{\downarrow}(\lambda)$ due to cloud changes has an immediate effect on $\rho(\lambda)$ that is calculated with Eq. 1. For illustration, Fig. 4 shows $\rho(\lambda)$ of a vegetated surface for an intermediate value $\theta$ of $45°$ and for four different combinations of $\tau_{\mathrm{cal}}$ and $\tau_{\mathrm{mea}}$. The ground truth $\rho(\lambda)$, as it would be obtained from satellites, with $\tau_{\mathrm{cal}} = \tau_{\mathrm{mea}} = 0$, is given by the black line. Only in the trivial case, when the airborne observations are performed under the same cloud-free conditions, the same value of $\rho(\lambda)$ would be measured. For reference, $\rho(\lambda)$ under constant cloud conditions during RP calibration and measurement, where

$\tau_{\mathrm{cal}} = \tau_{\mathrm{mea}} \neq 0$, is indicated by the gray line.

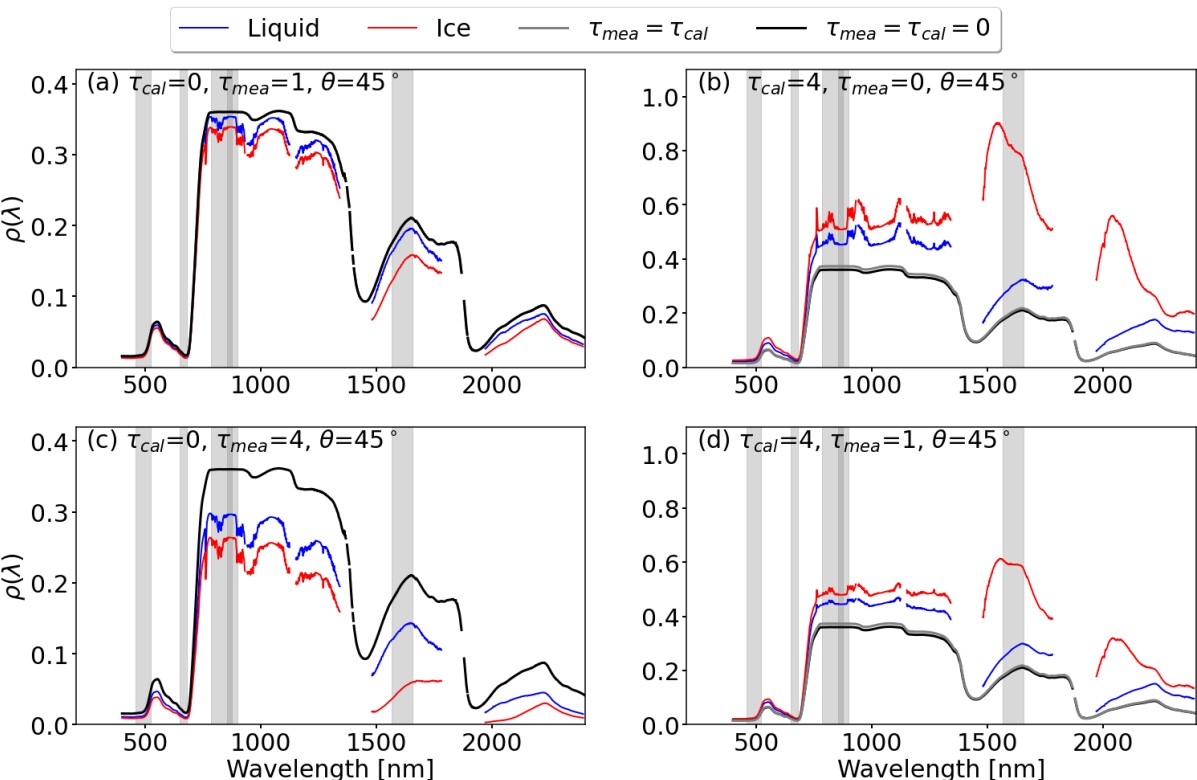

**Figure 4.** Panels **(a)** and **(c)**: Synthetic spectral reflectance $\rho(\lambda)$ measurements, when cloud conditions change from $\tau_{\mathrm{cal}} = 0$ to $\tau_{\mathrm{mea}} = 1$ and 4, respectively. Panels **(b)** and **(d)**: Same as **(a)** and **(b)** but for $\tau_{\mathrm{cal}} = 4$ and values of $\tau_{\mathrm{mea}}$ of 1 and 4, respectively. In all panels: The black line represents $\rho(\lambda)$ under cloud-free conditions ($\tau_{\mathrm{cal}} = \tau_{\mathrm{mea}} = 0$) and the gray line represents $\rho(\lambda)$ under constant cloudy conditions ($\tau_{\mathrm{cal}} = \tau_{\mathrm{mea}} \neq 0$). Red lines represent ice water clouds and blue lines represent liquid water clouds. The gray marked areas highlight the Sentinel-2 bands B2, B4, B8, B8a, and B11.

The cases shown in Fig. 4a and c represent cloud-free conditions during the RP calibration ($\tau_{\mathrm{cal}} = 0$), while clouds are present during the actual measurements with $\tau_{\mathrm{mea}} = 1$ and $\tau_{\mathrm{mea}} = 4$, respectively. Due to the cloud change with $\tau_{\mathrm{cal}} > \tau_{\mathrm{mea}}$, the amused $\rho(\lambda)$ is lower compared to the reference. The difference between measured $\rho(\lambda)$ and the reference increases with increasing difference $\Delta\tau = \tau_{\mathrm{mea}} - \tau_{\mathrm{cal}}$ and is more pronounced for ice clouds than for liquid water clouds of the same $\tau$. For the ice cloud, $\rho(\lambda)$ at about 842 nm wavelength (Sentinel-2 B8) is reduced from about 0.38 ($\tau_{\mathrm{cal}} = \tau_{\mathrm{mea}} = 0$) to about 0.33 ($\tau_{\mathrm{cal}} = 0, \tau_{\mathrm{mea}} = 1$) and 0.26 ($\tau_{\mathrm{cal}} = 0, \tau_{\mathrm{mea}} = 4$). Similarly, the opposite situation is possible, where $\tau$ decreases after calibration ($\tau_{\mathrm{cal}} > \tau_{\mathrm{mea}}$), which is shown in the right column in Fig. 4. Since $\tau_{\mathrm{cal}} > \tau_{\mathrm{mea}}$, the measured $\rho(\lambda)$ overestimates the expected ground truth $\rho(\lambda)$, with the greatest bias in $\rho(\lambda)$ at about 1610 nm wavelength (Sentinel-2 B10) in the case of an ice cloud.

The examples given in Fig. 4b and d show that the estimated $\rho(\lambda)$ under constant cloud conditions (gray lines) are slightly enhanced compared to $\rho(\lambda)$ under cloud-free conditions but the change in $\rho(\lambda)$ due to differences between $\tau_{\mathrm{cal}}$ and $\tau_{\mathrm{mea}}$ are

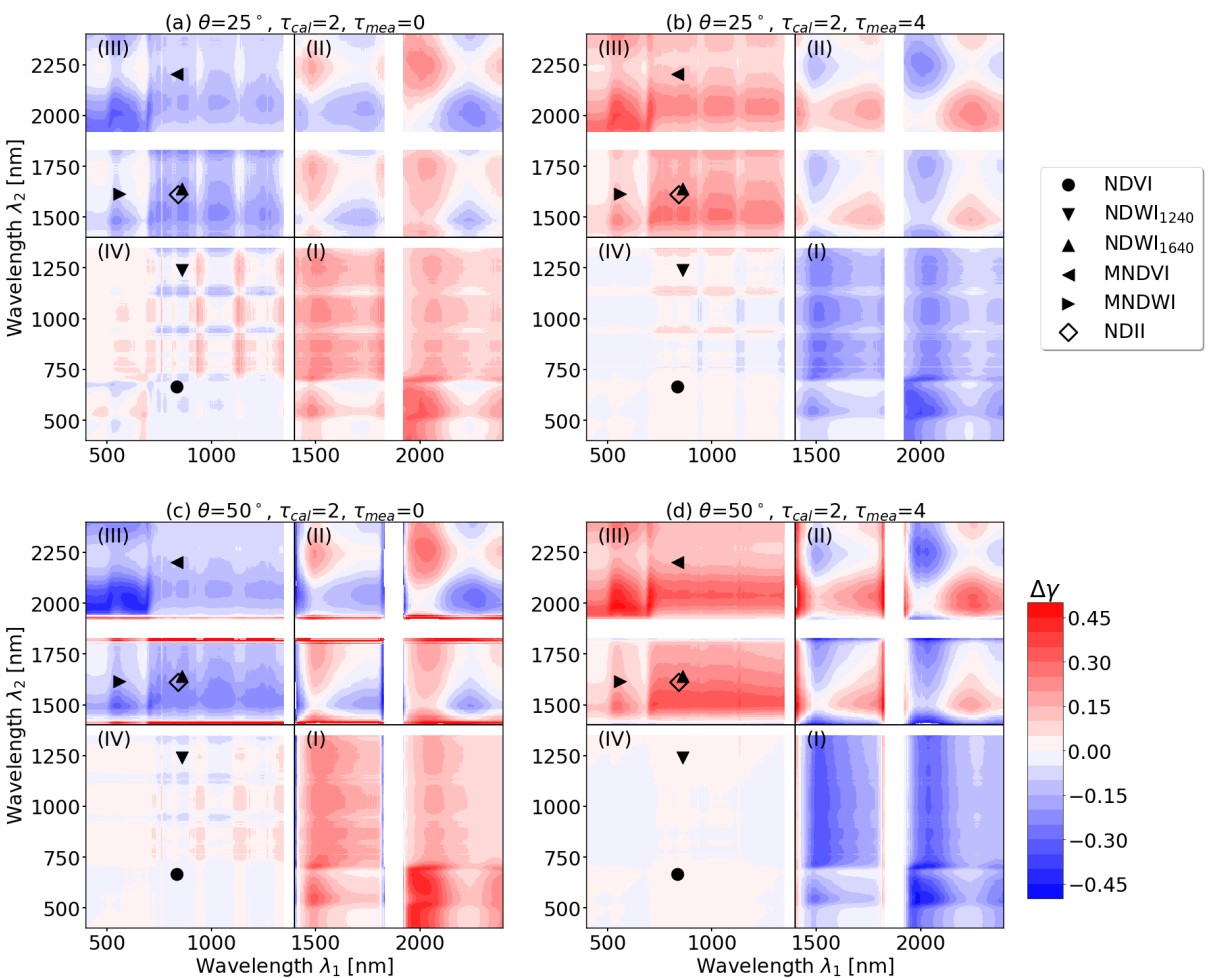

**Figure 5.** Absolute difference $\Delta\gamma = \gamma_{\mathrm{mea}} - \gamma$ for change in illumination conditions, when cloud optical thickness is different during the reflectance panel measurement ($\tau_{\mathrm{cal}}$) and actual condition during the measurement ($\tau_{\mathrm{mea}}$). Absolute differences in $\gamma$ are given for combinations of $\lambda_1$ ($x$-axis) and $\lambda_2$ ($y$-axis) simulated for an ice cloud, and for solar zenith angles $\theta$ of $25°$ (top) and $50°$ (bottom). Left column: Absolute differences, when $\tau_{\mathrm{cal}} = 2$ and $\tau_{\mathrm{mea}} = 0$. Right column: Absolute differences, when $\tau_{\mathrm{cal}} = 2$ and $\tau_{\mathrm{mea}} = 4$. Red areas show an overestimation of $\gamma$, while blue areas indicate an underestimation of $\gamma$. Wavelength combinations of $\gamma$ affected by strong water vapor absorption between 1350 and 1400 nm as well as 1830 and 1920 nm have been masked.

much greater and dominating. Therefore, the subsequent analysis primarily focuses on the contribution of changing cloud conditions.

## 3.2 Effect of clouds and cloud changes on two-band vegetation indices

Figure 4 showed that the spectral distortion due to differences in $\tau_{\mathrm{cal}}$ and $\tau_{\mathrm{mea}}$ affects certain wavelengths stronger than others. Subsequently, all wavelength combinations of $\rho(\lambda_1)$ and $\rho(\lambda_2)$, with $\lambda_1, \lambda_2 \in [400, 2400]$, that might be used in two-band VIs following Eq. 8 are examined with respect to their sensitivity to cloud changes. A similar approach was used, for example, by Werner et al. (2013) to determine the effect of cirrus clouds on upward radiances used for cloud remote sensing.

Figure 5 shows the effect of cloud changes on $\gamma$ expressed as $\Delta\gamma = \gamma_{\mathrm{mea}} - \gamma$, with $\gamma_{\mathrm{mea}}$ obtained over vegetation using the latest RP calibration that was performed under $\tau_{\mathrm{cal}}$. The expected ground truth $\gamma$ is the value that would be expected if an immediate estimate of $\kappa$ would be available or cloud conditions would not have changed ($\tau_{\mathrm{cal}} = \tau_{\mathrm{mea}} \neq 0$). A negative $\Delta\gamma$ therefore indicates an underestimation of the true $\gamma$ and vice versa. The combinations of $\tau_{\mathrm{cal}}$ and $\tau_{\mathrm{mea}}$ were chosen to represent optically thin cirrus with changes in $\tau$ that could occur between RP calibration measurements, e.g., 10 minutes apart. Each panel in Fig. 5 is divided into four quadrants, starting with the first quadrant (Q1) in the lower right part and turning counter-clockwise. It follows from Eq. 8 that $\Delta\gamma$ is point-symmetric with respect to the diagonal from the origin to the upper right corner. Thus, for any combination of $\lambda_1$ and $\lambda_2$ that causes an overestimation or underestimation, the inverse wavelength combination $\lambda_2$ and $\lambda_1$ will cause the same bias $\Delta\gamma$, but with opposite sign. In addition to NDVI and NDWI, selected two-band VIs that follow Eq. 8 and use SWIR wavelengths were added to the plot, namely: the modified normalized difference water index (MNDVI; Jurgens, 1997), the modified normalized difference water index (MNDWI; Xu, 2006), and the normalized difference infrared index (NDII; Hardisky et al., 1983). The list is not comprehensive and many more VIs exist, e.g., tabulated by Jones et al. (2012), Zeng et al. (2022), and Montero et al. (2023). Wavelengths affected by strong water vapor absorption between 1350 and 1400 nm as well as 1830 and 1920 nm wavelengths have been masked because $F^\downarrow(\lambda)$ and $F^\uparrow(\lambda)$ are close to zero causing numerical unstable values of $\rho(\lambda)$.

The largest absolute values of $|\Delta\gamma|$, indicated by dark red or dark blue colors, occur for wavelength combinations with the greatest spectral distance $|\Delta\lambda| = |\lambda_1 - \lambda_2|$. Also wavelength combinations with one wavelength greater than 1400 nm, i.e., in Q1 and Q3, are affected by cloud transitions, since the spectral slope in $F^\downarrow(\lambda)$ is most pronounced towards longer wavelengths (see Fig. 3).

In contrast, independent of the selected combination of $\tau_{\mathrm{cal}}$, $\tau_{\mathrm{mea}}$, and $\theta$ generally low values of $|\Delta\gamma|$ are found for wavelength combinations with $\lambda < 1400$ nm (Q4). Additionally, small values of $\Delta\gamma$ also occur along the diagonal from the origin to the upper right corner, i.e., the smaller $|\Delta\lambda|$ becomes. While small $|\Delta\lambda|$ would minimize the cloud effect, the proximity of $\lambda_1$ and $\lambda_2$ limits the information content that can be extracted from spectral ratios. A trade-off between information content and small cloud influence is required and Fig. 5 provides guidance to choose suitable wavelength combinations.

Figure 5a illustrates $\Delta\gamma$ for $\theta = 25°$, where $\tau_{\mathrm{cal}} = 2$ decreases to $\tau_{\mathrm{mea}} = 0$. The decrease in $\tau$ after the RP calibration leads to predominantly negative $\Delta\gamma$ (blue colors) in Q3 and positive $\Delta\gamma$ (red colors) in Q2, indicating a respective underestimation and overestimation of the true $\gamma$. In the selected case, $\Delta\gamma$ of NDVI (black circle) is small compared to the full potential range of NDVI (see Table 3). Greater values of $\Delta\gamma$ are calculated for $\mathrm{NDWI}_{1640}$ with up to 0.184, resulting from the second wavelength being located in the SWIR region that is subject to the spectral slope in $F^\downarrow(\lambda)$.

**Table 3.** Difference $\Delta\gamma = \gamma_{\mathrm{mea}} - \gamma$ between estimated vegetation index $\gamma_{\mathrm{mea}}$ and ground truth vegetation index $\gamma$ for four vegetation indices for the four example cases given in Fig. 5a–d. The calculations assume a spherical LAD. Sentinel-2 band ratios were taken from Montero et al. (2023).

| Vegetation index | Sentinel-2 bands ratios | $\Delta\gamma$ | | | |
|---|---|---|---|---|---|
| | | Case (a) | Case (b) | Case (c) | Case (d) |
| NDVI | (B8−B4) / (B8+B4) | 0.043 | −0.011 | 0.009 | −0.007 |
| $\mathrm{NDWI}_{1240}$ | (B3−B8) / (B3+B8) | −0.048 | 0.018 | −0.022 | 0.015 |
| $\mathrm{NDWI}_{1640}$ | (B8a−B11) / (B8a+B11) | 0.184 | −0.212 | 0.216 | −0.267 |
| MNDVI | (B8−B12) / (B8+B12) | 0.111 | −0.127 | 0.112 | −0.162 |
| MNDWI | (B3−B11) / (B3+B11) | 0.08 | −0.113 | 0.156 | −0.148 |
| NDII | (B8−B11) / (B8+B11) | 0.173 | −0.205 | 0.208 | −0.264 |

The second example in Fig. 5b shows the opposite transition from an optically thin cloud during the RP calibration with $\tau_{\mathrm{cal}} = 2$ to an optically thicker cloud during measurement $\tau_{\mathrm{mea}} = 4$. This leads to an inverted pattern of $\Delta\gamma$. Although the change in cloudiness $|\Delta\tau| = \tau_{\mathrm{cal}} - \tau_{\mathrm{mea}}$ is similar to the example in Fig. 5a, the magnitude of $|\Delta\gamma|$ for all wavelength combinations in Q2 and Q4 are slightly smaller, while the effect is greater for wavelength combinations in Q1 and Q3. This shows that the bias $|\Delta\gamma|$ from changing cloud conditions is affected by $\Delta\tau$ but also depends on the absolute value of $\tau_{\mathrm{cal}}$ during calibration. For this example, NDVI and $\mathrm{NDWI}_{1640}$ are subject to biases $\Delta\gamma$ of 0.011 and −0.212, respectively.

Figure 5c shows $\Delta\gamma$ for the same combination of $\tau_{\mathrm{cal}}$ and $\tau_{\mathrm{mea}}$ that is given in Figure 5a but for a greater value of $\theta$ of $50°$. This leads to greater absolute values of $|\Delta\gamma|$ in Q1–Q3. An exception is Q4, which is characterized by $\Delta\gamma$ around 0 for all wavelengths with less pronounced water vapor absorption features compared to $\theta$ of $25°$. This indicates that with increasing $\theta$, changes in $\tau$ lead to reduced biases in $\gamma$ for wavelengths less than 1400 nm, while larger biases in $\gamma$ are expected for VIs using wavelengths beyond 1400 nm. In general, the effects of changing cloud conditions on $\rho(\lambda)$ and $\Delta\gamma$ are less pronounced for the planophile LAD, while for the erectophile LAD greater effects were determined. This results primarily from the higher sensitivity of $\rho(\lambda)$ on $\theta$ in case of the erectophile LAD (Wolf et al., 2025a). An overview of $\Delta\gamma$ for all example two-band VIs derived for the four cases that are marked in Fig. 5 are given in Table 3.

Subsequently, the effects of changing $\tau$ between RP calibration and measurement on three selected VIs are investigated.

### 3.2.1 Effect of cloud changes on the normalized differential vegetation index (NDVI)

The estimated NDVI for three values of $\theta$ depending on the combination of $\tau_{\mathrm{cal}}$ and $\tau_{\mathrm{mea}}$ is shown in the top row of Fig. 6.

First, we consider the trivial case of cloud-free conditions with $\tau_{\mathrm{cal}} = \tau_{\mathrm{mea}} \approx 0$, where NDVI of about 0.87, 0.9, and 0.92 are calculated for $\theta$ of $25°$, $50°$, and $70°$, respectively. These cases are marked by the dark-blue dots and represent the reference NDVI. The increase of NDVI with increasing $\theta$ is related to scattering and absorption at gas molecules and aerosol particles. Next, cloudy conditions are considered, where $\tau_{\mathrm{cal}} \approx \tau_{\mathrm{mea}} \neq 0$ (colored dots), showing that the NDVI is enhanced by the

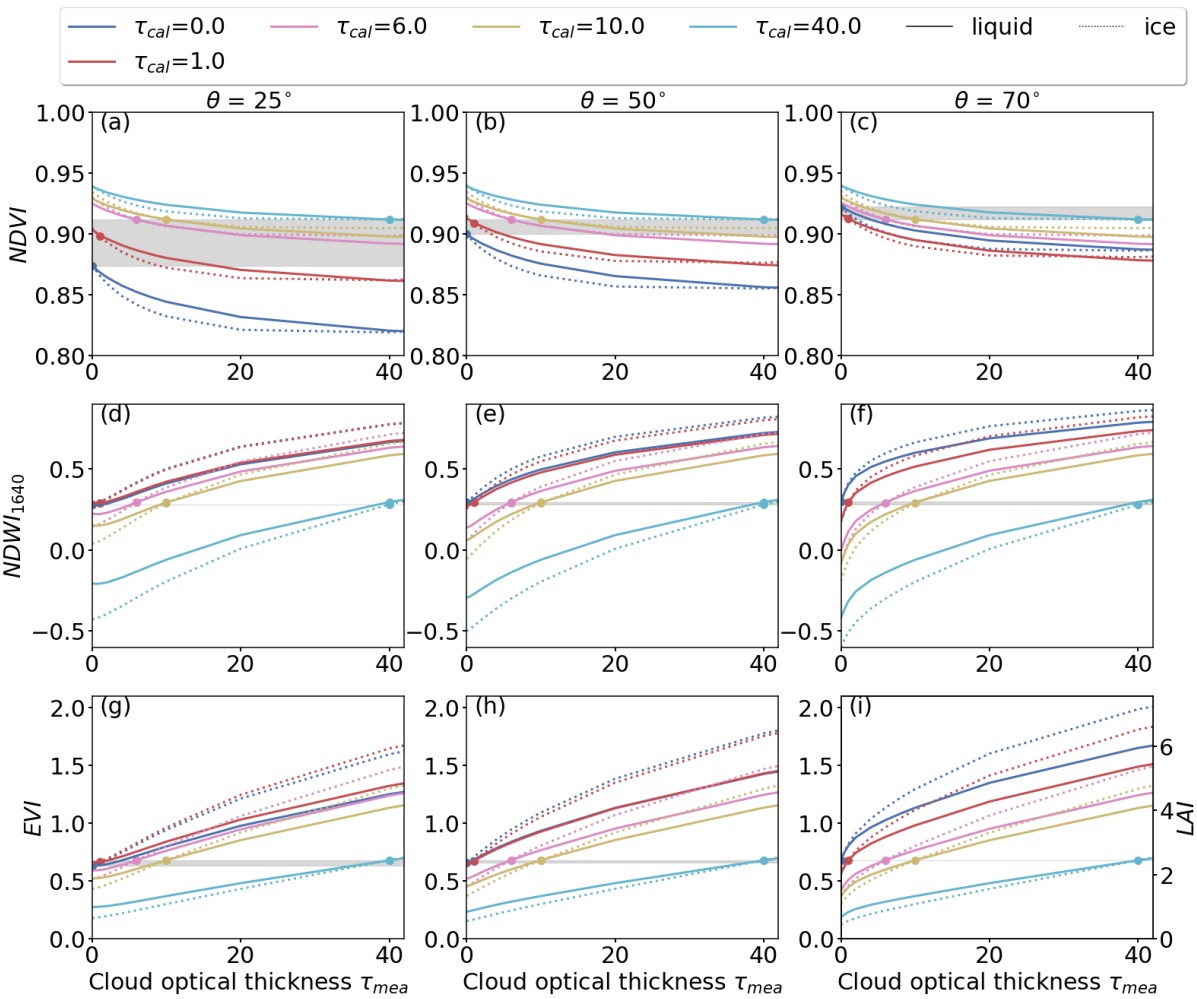

**Figure 6.** Top row: Absolute values of NDVI as obtained for solar zenith angles $\theta$ of 25°, 50°, and 70°. Subsequent rows same as top row but for $\mathrm{NDWI}_{1640}$ and EVI. Color-coded is the cloud optical thickness $\tau_{\mathrm{cal}}$ that was present during the reflectance panel measurement. The cloud optical thickness $\tau_{\mathrm{mea}}$ during the measurement is given on the $x$-axis. Simulations for the liquid water cloud are given by solid lines and simulations for the ice cloud are given by the dotted lines. All simulations base on the assumption of a spherical LAD. The variability in VI due to the presence of clouds ($\tau_{\mathrm{cal}} = \tau_{\mathrm{mea}}$) and the associated change in the surface reflectance, is highlighted in gray.

presence of the cloud (gray highlighted area), which influences $f_{\mathrm{dir}}(\lambda)$ and $\rho(\lambda)$. The greatest variability is found for $\theta = 25°$, where NDVI increases from 0.87 to 0.91 with increasing $\tau$, which could be interpreted as an overestimation of vegetation health. Smaller effects are found for $\theta = 50°$ and for $\theta = 70°$ the NDVI even decreases. It should be noted that measuring NDVI under cloud-free conditions at different times of day, i.e., different $\theta$, causes the same variability as measuring NDVI at a fixed time with $\theta = 25°$ but under different cloud conditions, for example, on consecutive days.

For cases where cloud conditions differ between $\tau_{\mathrm{cal}}$ and $\tau_{\mathrm{mea}}$, the change of $\tau_{\mathrm{mea}}$ for fixed $\tau_{\mathrm{cal}}$ is given by the colored lines. Moving to the right along lines of same $\tau_{\mathrm{cal}}$ can be understood as the advection of an optically thicker cloud during the measurement, while moving left represents the advection of an optically thinner cloud. The largest differences between measured and reference NDVI generally occur for $\theta = 25°$, and become successively smaller with increasing $\theta$. Thus, the NDVI is subject to the largest biases from cloud transitions when $\theta$ is small. The advection of an optically thicker cloud after the RP measurement ($\tau_{\mathrm{cal}} < \tau_{\mathrm{mea}}$) results in a decrease in NDVI and an underestimation of the expected value. For $\tau_{\mathrm{cal}} = 0$ an extreme increase of $\tau_{\mathrm{mea}}$ from 0 to 40 results in a decrease in NDVI from 0.87 to 0.84, which could be interpreted as an underestimation of vegetation health. The advection of an optically thinner cloud after the RP measurement results in an overestimation of the actual NDVI value. For example, the combination of $\tau_{\mathrm{cal}} = 40$ and $\tau_{\mathrm{mea}} = 0$ increases the NDVI from 0.92 to 0.94. The aforementioned extreme changes in $\tau$ from 0 to 40, and vice versa, between RP overflights are unlikely in the case of stratiform clouds. These extremes of $\tau$ have been selected to estimate the maximum envelope for biases in NDVI. Biases in NDVI, retrieved under more homogeneous conditions, will be smaller. However, even seemingly stratiform clouds vary in $\tau$ and therefore bias the retrieved values of NDVI. These variations have the greatest impact, when the RP overflight is performed under cloud-free conditions ($\tau_{\mathrm{cal}} = 0$) or under conditions with small values of $\tau_{\mathrm{cal}}$, since the slope of the curves is greatest in these situations (see, for example, the blue lines in Fig. 6a–c).

Comparing the responses of NDVI obtained below liquid and ice water clouds shows that both cloud types cause similar biases. However, the magnitude is generally greater for the ice cloud with the largest difference for small values of $\theta$ and for $\tau_{\mathrm{mea}}$ between 10 and 30. However, it is acknowledged that the calculated deviations between the measured and expected NDVI are small, considering the selected differences between $\tau_{\mathrm{cal}}$ and $\tau_{\mathrm{mea}}$, and the typical range of NDVI between 0 and 1.

### 3.2.2 Effect of cloud changes on the normalized differential water index (NDWI)

Remote sensing of $\mathrm{NDWI}_{1640}$ is based on the Sentinel-2 bands B8a and B11, a combination that is more sensitive to changes in $\tau$ compared to the NDVI, because one wavelength is located in the SWIR (Q3 in Fig. 5). The middle row in Fig. 6 shows the change of NDWI induced by cloud changes.

Regardless of $\theta$, the bias in $\mathrm{NDWI}_{1640}$ is under cloudy conditions with $\tau_{\mathrm{cal}} = \tau_{\mathrm{mea}}$ is of similar magnitude compared to the NDVI. A variation in $\mathrm{NDWI}_{1640}$ with maximal values of $\pm 0.02$ (gray highlighted area) is calculated, which is small compared to the influence of changes in cloudiness between RP calibrations. Here, the change in cloud conditions is discussed for $\theta = 25°$. When the RP measurements was performed for $\tau_{\mathrm{cal}} = 0$ the advection of an optically thicker cloud, represented by an increase in $\tau_{\mathrm{cal}}$ from 0 to 40, leads to an increase in $\mathrm{NDWI}_{1640}$ from 0.28 to 0.89 ($\Delta\gamma = 0.61$), which is relevant considering the typical range between $-1$ and 1. In such scenarios, the interpreted true health status would be well overestimated using

NDWI$_{1640}$. For the same $\theta$, a calibration under cloudy sky with $\tau_{\mathrm{cal}} = 40$ and a subsequent decrease in $\tau_{\mathrm{mea}}$ from 40 to 0 results in a decrease in NDWI$_{1640}$ from 0.59 to $-0.43$ ($\Delta\gamma = -1.02$). Even though $\Delta\tau$ is equal in both cases, the resulting

$\Delta\gamma$ differs, which emphasizes the dependence of the bias $\Delta\gamma$ on the absolute $\tau_{\mathrm{cal}}$. As for the NDVI, the transitions in $\tau$ from 0 to 40, or vice versa, are extreme cases. For practical applications, variations in $\tau_{\mathrm{mea}}$ for a given $\tau_{\mathrm{cal}}$ are more relevant. For the NDWI$_{1640}$, the slopes in Fig. 6d–f are shifted in absolute terms but are constant for the same values of $\theta$. Therefore, variations in $\tau_{\mathrm{mea}}$ between RP overflights will cause the same bias in NDWI$_{1640}$, independent of $\tau_{\mathrm{cal}}$. Similar to NDVI, the effect of ice clouds on NDWI$_{1640}$ is qualitatively similar to the effect from liquid water clouds, however the magnitude is almost twice as

large.

### 3.3    Effect of clouds and cloud changes on the three-band enhanced vegetation index (EVI)

As an example for a three-band VI, the effects of changing cloud conditions between RP calibration and measurement are shown for the EVI (see Fig. 6g–i). The effect of changes in $f_{\mathrm{dir}}(\lambda)$ under constant conditions ($\tau_{\mathrm{cal}} = \tau_{\mathrm{mea}}$) is generally small (gray area) compared to the variation in EVI that is associated with the mismatch between $\tau_{\mathrm{cal}}$ and $\tau_{\mathrm{mea}}$. Irrespective of $\theta$, the

advection of an optically thicker cloud ($\tau_{\mathrm{cal}} < \tau_{\mathrm{mea}}$) causes an increase in estimated EVI, which leads to an overestimation of the true EVI. Conversely, the advection of an optically thinner cloud ($\tau_{\mathrm{cal}} > \tau_{\mathrm{mea}}$) leads to and underestimation of the true EVI. The bias gets more pronounced with increasing difference between $\tau_{\mathrm{cal}}$ and $\tau_{\mathrm{mea}}$, and with increasing absolute value of $\tau_{\mathrm{cal}}$. The magnitude of these biases are more pronounced for the ice cloud than for the liquid water cloud. Furthermore, the biases become larger with increasing $\theta$, as well as with increasing difference between $\tau_{\mathrm{cal}}$ and $\tau_{\mathrm{mea}}$, which is particularly pronounced

in combination with small $\tau_{\mathrm{cal}}$. Thus, estimated EVI are less susceptible to changes in cloud conditions at low values of $\theta$, e.g., around noon or at low latitudes, compared to cloud changes during measurements with the Sun close to the horizon or generally at higher latitudes. As an example, a calibration performed under $\theta = 25°$ and cloud-free conditions followed by the advection of an optically thicker ice cloud with $\tau_{\mathrm{mea}} = 4$ results in an increase in EVI from the expected value of 0.67 to 0.75. For the same transition in $\tau$ but for $\theta = 70°$ an increase from 0.67 to 1.04 is estimated. In both cases, the inferred ground-truth

vegetation health would be overestimated.

The different response of EVI to cloud changes compared to the NDVI is related to two factors. First, the equation to calculate the EVI is fundamentally different from the two-band VIs. The use of an additional third spectral band at 490 nm wavelength makes the EVI generally more responsive to the spectral slope in $F^{\downarrow}(\lambda)$ caused by absorption. Second, the spectral slope effect is amplified by the pre-factors in Eq. 9, since re-writing Eq. 9 results in:

$$\mathrm{EVI} = G \cdot \frac{a - b}{a + C_1 \cdot b - C_2 + L/\rho_{\mathrm{B2}}}, \tag{10}$$

with the reflectance ratios $a = \rho_{\mathrm{B8}}/\rho_{\mathrm{B2}}$ and $b = \rho_{\mathrm{B4}}/\rho_{\mathrm{B2}}$ at Sentinel-2 bands B2, B4, and B8 (see Table 1). The ratios $a$ and $b$ are affected by the spectral slope in the illumination ratio (see Fig. 3). The spectral slope is greater for $a$ than for $b$, since $\Delta\lambda$ is greater in $a$ than in $b$. The differences between $a$ and $b$ do not cancel out and are amplified by the pre-factor $C_1$. The term $L/\rho_{\mathrm{B2}}$ also contributes, since the constant $L$ is inversely scaled with $\rho_{\mathrm{B2}}$ and band B2 is sensitive to atmospheric scattering.

## 3.4 Implication of biases in vegetation indices on estimated biophysical properties

In vegetation remote sensing, VIs are used to estimate biophysical properties such as LAI, gross primary production (GPP), fresh and dry biomass, and vegetation water content (Gitelson et al., 2006; Hong et al., 2007; Ahmadian et al., 2016). For example, Boegh et al. (2002) proposed an empirical linear regression given by:

$$\text{LAI} = 3.618 \cdot \text{EVI} - 0.118, \tag{11}$$

where the LAI scales linearly with the EVI. Due to the direct relationship between EVI and LAI, biases in EVI are linearly scaled by the pre-factor 3.618 and lead to biased estimates of LAI. For example, when a calibration for $\theta = 50°$ was performed under cloud conditions with $\tau_{\text{cal}} = 0$ but the measurement is influenced by an ice cloud with $\tau_{\text{mea}} = 10$. This results in a bias in EVI of 0.25, which corresponds to a bias in LAI of about $\Delta\text{LAI} = 0.25 \cdot 3.618 = 0.9$.

Similarly, the gross primary product is an important measure in the context of the global carbon cycle, since it indicates how quickly an ecosystem accumulates biomass (Gitelson et al., 2006). Several attempts have been made to estimate GPP based on NDVI and EVI; for example Rahman et al. (2005), Wu et al. (2009), or Zhou et al. (2014). Correlations have been estimated using in-situ carbon uptake flux measurements and remotely sensed NDVI or EVI. Although the correlations between GPP and EVI or NDVI vary by site and crop type, stable correlations could be identified. Using the correlations from Rahman et al. (2005) or Zhou et al. (2014), a variation in EVI of $\Delta\text{EVI} = 0.2$ would yield variations in estimated GPP of about $2.6 \text{ gC} \text{m}^{-2} \text{d}^{-1}$. Zhou et al. (2014) also provided relationships for NDVI, where a variation of $\Delta\text{NDVI} = 0.2$ would result in a variation in GPP of about $0.51 \text{ gC} \text{m}^{-2} \text{d}^{-1}$. Considering that the total range of GPP spans from $0 \text{ gC} \text{m}^{-2} \text{d}^{-1}$, when the NDVI or EVI is zero, to $8 \text{ gC} \text{m}^{-2} \text{d}^{-1}$, when NDVI or EVI is close to one, the bias in estimated GPP from biases in EVI is of relevance.

Several studies have attempted to estimate vegetation water content (VegWC) based on NDVI (Hong et al., 2007). The relationships derived between NDVI and VegWC vary greatly depending on the year, location, and crop type. Therefore, the identified cloud-induced biases in NDVI are expected to play only a minor role in the total uncertainty of estimated VegWC. Ahmadian et al. (2016) aimed to estimate fresh and dry biomass, and LAI based on derived NDVI and EVI. These relationships are strongly dependent on vegetation type. Therefore, cloud-induced variations in NDVI are expected to contribute only a small amount to the total uncertainty in estimated fresh and dry biomass. However, since EVI is more sensitive to cloud-induced biases than NDVI, the uncertainty in EVI can be of a similar order of magnitude compared to the uncertainty in the relationships.

## 4 Summary and conclusions

This paper presented results of coupled atmosphere-vegetation radiative transfer (RT) simulations, using the coupled RT models libRadtran and SCOPE2.0, to systematically investigate biases in remotely sensed vegetation indices (VIs) due to changing cloud conditions. Simulations were performed for a stratiform liquid water cloud and an ice cloud representative of high-level cirrus. The optical thickness of the cloud $\tau$ varied between 0 and 40, and the solar zenith angles $\theta$ ranged from 25°

to 70°. The optical properties were represented by a spherical leaf angle distribution (LAD) and a leaf area index (LAI) of 3. The simulations were designed to resemble below-cloud measurements of downward irradiance $F^\downarrow(\lambda)$ and spectral surface reflectance $\rho(\lambda)$ used for remote sensing of vegetation from airborne observations. The synthetic measurements mimic the typical observation strategy during field measurements, where reflectance panels (RPs) are used to calibrate the surface reflectance measurements.

Field measurements can be performed below clouds, which reduce the direct fraction $f_{\mathrm{dir}}(\lambda)$ in $F^\downarrow(\lambda)$, which is a controlling factor of the surface reflectance. Clouds also change the downward irradiance $F^\downarrow(\lambda)$ spectrally and in absolute terms through spectral dependent scattering and absorption. Changes in cloud conditions, expressed as cloud optical thickness $\tau$, between periodic RP calibrations are expected to cause spectral distortions in the estimated $\rho(\lambda)$ and thus introduce biases in VIs estimated in the presence of clouds.

The synthetic observations allowed to separate the effect from changes in $f_{\mathrm{dir}}(\lambda)$ from the changes in $\tau$ between RP calibrations. For a solar zenith angle $\theta = 25°$, a reduction in $f_{\mathrm{dir}}(\lambda)$ led to an increase in $\rho(\lambda)$ for wavelengths greater than 650 nm compared to cloud-free conditions, while for wavelengths below 650 nm $\rho(\lambda)$ was reduced. For $\theta > 55°$, $\rho(\lambda)$ below 650 nm wavelengths was also increased with decreasing $f_{\mathrm{dir}}(\lambda)$. The change in $\rho(\lambda)$ under cloudy conditions therefore depends on the combination of $\theta$ and $\tau$, and the resulting $f_{\mathrm{dir}}(\lambda)$. The effect of changes in $f_{\mathrm{dir}}(\lambda)$ on estimated VIs was found to be small compared to the effect of changes in $\tau$ between RP calibrations.

The influence of cloud changes on two-band VIs was investigated for the normalized difference vegetation index (NDVI) and the normalized water index (NDWI). Other examples of two-band VIs were provided. The influence of cloud changes on the enhanced vegetation index (EVI), which is representative for three-band VIs, was also investigated. The effect of cloud changes on narrow-band VIs was found to be small for VIs using wavelength combinations below 1400 nm and for decreasing spectral distance between wavelength combinations. An optimum must be found between the information content obtained from the wavelength ratio and the sensitivity of the respective wavelengths pair to cloud changes. Guidance was provided by presenting the cloud-induced bias for potential two-band wavelength combinations between 400 and 2400 nm. For wavelengths greater than 1400 nm, the sensitivity to cloud changes was found to increase and is particularly pronounced in the proximity and within the water absorption bands. Wavelengths smaller than 1400 nm, were found to be less impacted. For the NDVI, a generally low sensitivity was found. For an intermediate value of $\theta = 50°$, the transition in $\tau$ from 0 to 10 led to a bias of about $-0.035$. With increasing $\theta$ and for same transition in $\tau$, the bias in NDVI increased, while a decrease in $\theta$ resulted in lower biases. The NDWI is subject to greater biases of up to 0.26 for an intermediate $\theta = 50°$ and a transition in $\tau$ from 0 to 10. The biases in NDWI generally increase with increasing $\theta$. The EVI was also found to be sensitive to changes in $\tau$. A transition in $\tau$ from 0 to 10 in combination with $\theta = 50°$ led to an overestimation of the EVI by 0.25. For the same transitions in $\tau$, the bias in EVI decreases with decreasing $\theta$. The leaf are index estimated from EVI using empirical equations is directly affected by potential biases in EVI. For example, the biases in EVI of 0.25 would cause a bias in LAI of 0.9. Similarly, biases in EVI and NDVI of about $\pm 0.2$ can lead to biases in estimated gross primary product of up to $\pm 2\,\mathrm{gCm^{-2}\,d^{-1}}$. The impact for other LADs, such as the erectophile and planophile, has been investigated. The erectophile LAD is more susceptible to biases in VIs under constant cloud conditions than the spherical LAD, while the bias in VIs with underlying planophile LAD is almost

unaffected. Although the absolute value of the VI biases are determined by the LAD, the relative change due to a transition in $\tau$ between RP measurements was found to be constant among the three LADs.

The presented analysis showed that the practice of using relative measurements with RPs is prone to uncertainties. With the improvement of drone technology and their ability to carry heavier payloads, and in combination with the advancements in sensor technology, it would be advantageous to measure $F^{\downarrow}(\lambda)$ directly instead of relying on relative measurements of $\rho(\lambda)$ using RPs. Uncertainties associated with changing cloud conditions could be minimized.

The simulations presented here are limited in their representation because the full natural variability in vegetation and canopy types could not be covered. Furthermore, the assumed default values used in the atmosphere and vegetation RT simulations can influence the results. In particular, the assumed LAI, LAD, plant dry matter, and soil properties in the vegetation RT simulations, and the assumed atmospheric profile in the atmosphere RT simulations can influence the results (Wolf et al., 2025a). Since the natural variability cannot be covered in a single study, the presented work is to be interpreted as a conceptual study and to highlight the potential impacts of clouds during field observations. It is also emphasized that the presented simulations are based on one-dimensional RT only and lack a more detailed representation of the three-dimensional nature of RT below heterogeneous cloud fields. This is particularly problematic in the vicinity of clouds, where a nearby cloud casts a shadow while no cloud is in the zenith. Nevertheless, we argue that the presented study can be used as a first approximation for the transition between cloud-free and cloudy regions.

*Data availability.* The simulated spectra of radiance, irradiance, and vegetation albedo are made available via NetCDF files. The data are available on the Zenodo platform via https://doi.org/10.5281/zenodo.15275610 (Wolf et al., 2025b).

## Appendix A: Effect of cloud changes on the NDVI, $\mathrm{NDWI_{1640}}$, and EVI for erectophile and planophile LADs

This section provides an overview of how an erectophile or planophile leaf angle distribution (LAD) affects biases in normalized differential vegetation index (NDVI), normalized difference water index ($\mathrm{NDWI_{1640}}$), and the enhanced vegetation index (EVI), compared to the spherical LAD presented in the main text. First, we consider constant cloud optical thickness during the reflectance panel (RP) overflight with $\tau_{\mathrm{cal}}$ and the actual measurement with $\tau_{\mathrm{mea}}$, such that $\tau_{\mathrm{cal}} = \tau_{\mathrm{mea}}$. Comparing Fig. 6 and Fig. A1 reveals a greater variation in the NDVI bias for the erectophile LAD than for the spherical LAD (gray highlighted area). The biases are greatest for small solar zenith angle $\theta$, when incoming radiation can penetrate deep into the canopy. In the case of $\theta = 25°$, the NDVI bias varies between 0.67 and 0.89 for the erectophile LAD, compared to a variation between 0.75 an 0.91 in case of the spherical LAD. In this particular case, the variation in NDVI exceeds the bias that results from a transition in cloud conditions, when $\tau_{\mathrm{cal}} \neq \tau_{\mathrm{mea}}$. However, when $\theta$ is greater than $25°$ and the penetration depth of the radiation into the canopy is reduced, the bias from changing cloud conditions between RP overflights again dominates the NDVI bias. Comparing Fig. 6 and Fig. A2 shows that for the planophile LAD and $\tau_{\mathrm{cal}} = \tau_{\mathrm{mea}}$, the variation in NDVI is negligible, with a

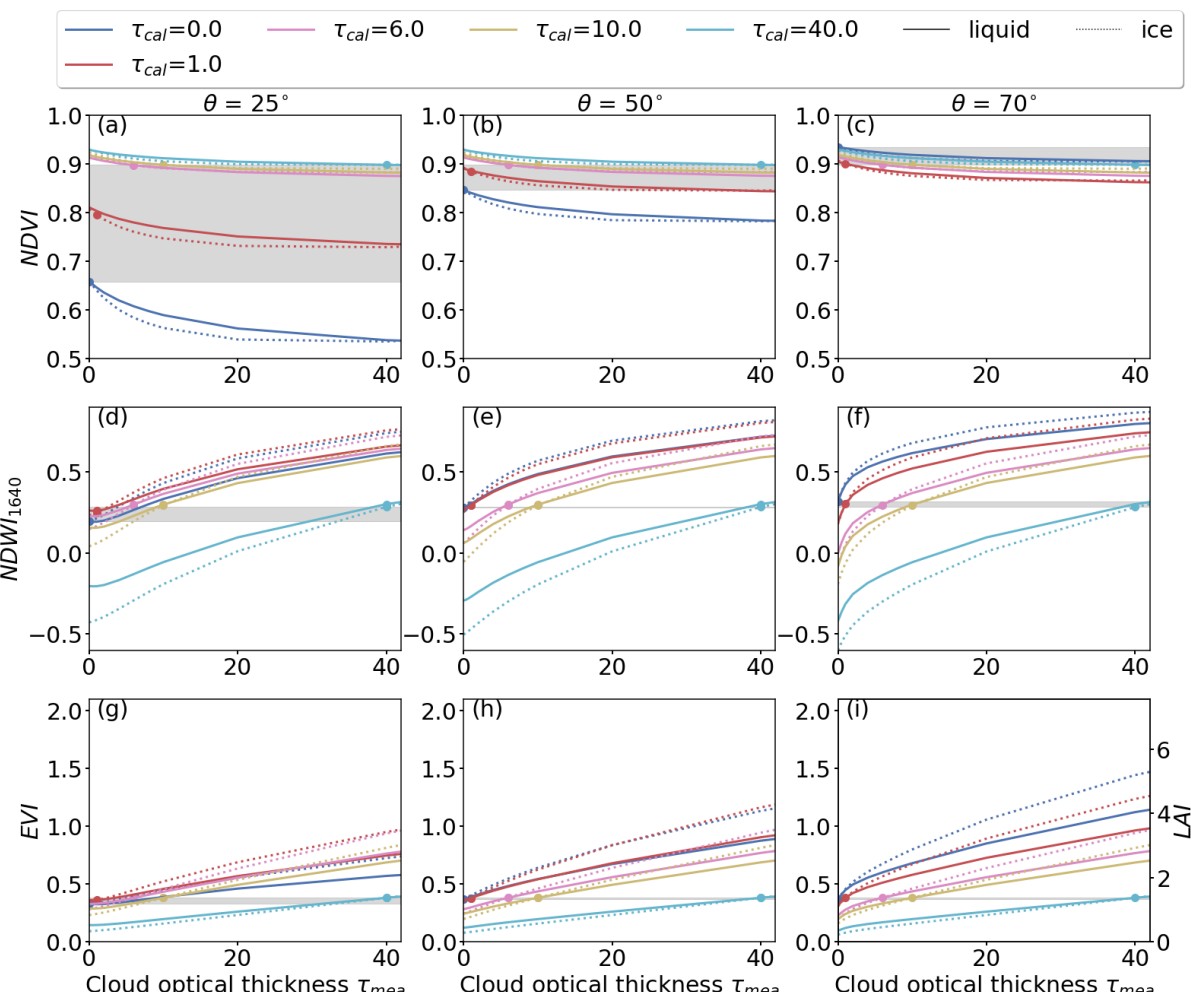

**Figure A1.** Same as Fig. 6 but for the erectophile leaf angle distribution.

nearly constant value of around 0.89. Similar effects are found for $NDWI_{1640}$ and EVI. In general, the greatest biases occur for the erectophile LAD, independent of $\theta$. Within on LAD the greatest biases occur for the lowest values of $\theta$.

Taking into account changes in cloud conditions, i.e., when $\tau_{cal} \neq \tau_{mea}$, the comparison of Fig. 6, Fig. A1, and Fig. A2 shows that for the same values of $\theta$, the relative variations in VI due to cloud changes are the same for all three LADs and VIs, which is indicated by a constant sensitivity of the VIs on $\tau_{mea}$ independent of the value of $\tau_{cal}$. However, the absolute values in the VIs are shifted along the $y$-axis depending on $\tau_{cal}$ and the selected LAD.

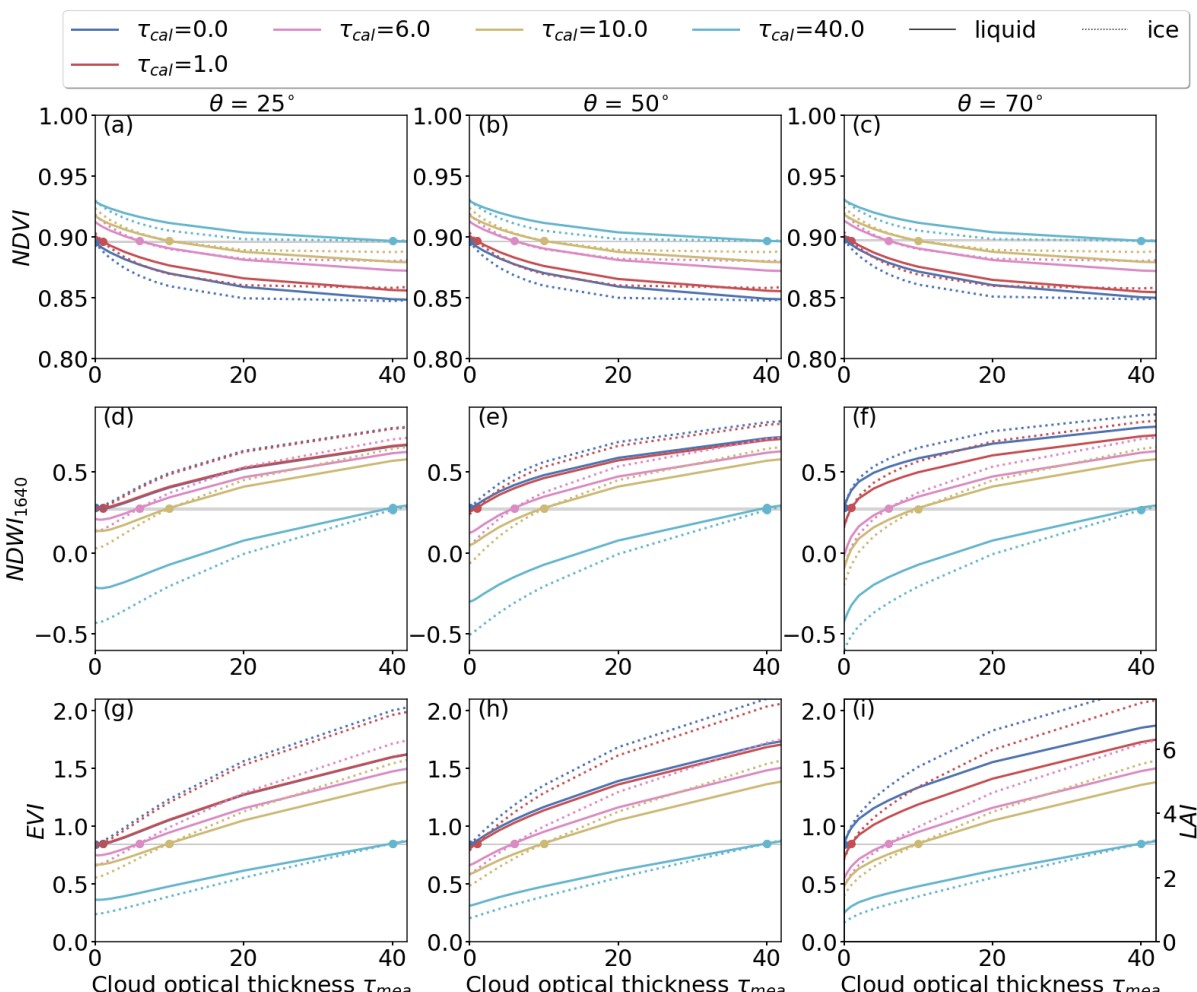

**Figure A2.** Same as Fig. 6 but for the planophile leaf angle distribution.

## Appendix B: Simulated reflectance factors and reflectance functions

The intrinsic reflectivity properties of a surface are given by its bidirectional reflectance distribution function (BRDF). The
495 BRDF quantifies the reflection and scattering of incident radiation on the surface from one direction of the hemisphere to
another. The spectral BRDF $f_{\mathrm{BRDF}}$, in units of $\mathrm{sr}^{-1}$, gives the ratio between the reflected radiance $I_{\mathrm{r}}^{\uparrow}(\theta_{\mathrm{i}}, \varphi_{\mathrm{i}}; \theta_{\mathrm{r}}, \varphi_{\mathrm{r}}; \lambda)$ with
respect to the incident irradiance $F_{\mathrm{i}}^{\downarrow}(\theta_{\mathrm{i}}, \varphi_{\mathrm{i}}; \lambda)$. The resulting $f_{\mathrm{BRDF}}$ depends on the zenith ($\theta_{\mathrm{i}}$) and the azimuth angle ($\varphi_{\mathrm{i}}$) of
the incoming radiation, and the zenith ($\theta_{\mathrm{r}}$) and azimuth angle ($\varphi_{\mathrm{r}}$) of the reflected radiation. The spectral BRDF $f_{\mathrm{BRDF}}$ also
depends on the wavelength $\lambda$ and is given by:

$$500 \quad f_{\mathrm{BRDF}} = \frac{\mathrm{d}I_{\mathrm{r}}^{\uparrow}(\theta_{\mathrm{i}}, \varphi_{\mathrm{i}}; \theta_{\mathrm{r}}, \varphi_{\mathrm{r}}; \lambda)}{\mathrm{d}F_{\mathrm{i}}^{\downarrow}(\theta_{\mathrm{i}}, \varphi_{\mathrm{i}}; \lambda)}, \tag{B1}$$

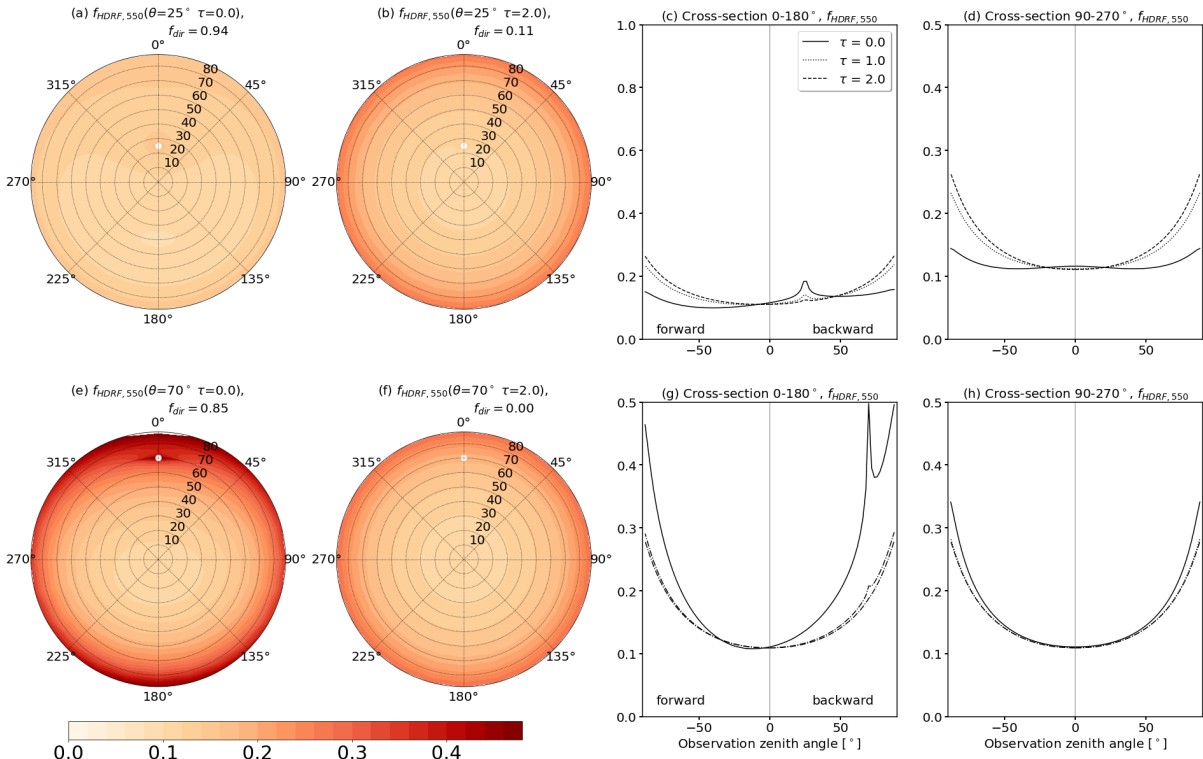

**Figure B1.** All sub-panels show the normalized hemispherical–directional reflectance factor $R_{\mathrm{HDRF}}$ at 550 nm wavelength. Panels **(a)**, **(b)**, **(e)**, and **(f)** present polar plots of $R_{\mathrm{HDRF}}$ for combinations of solar zenith angle $\theta$ and cloud optical thickness $\tau$. Panels **(c)**, **(d)**, **(g)**, and **(h)** present cross-sections of $R_{\mathrm{HDRF}}$ along lines of $0°$–$180°$ and $90°$–$270°$ azimuth.

given in the unit steradian ($\mathrm{sr}^{-1}$).

Under atmospheric conditions, where $F^{\downarrow}(\lambda)$ is composed of a direct and a diffuse fraction, the BRDF cannot be measured (Pinty et al., 2005; Schaepman-Strub et al., 2006). What can be observed is the spectral reflectance $\rho(\lambda)$ and the hemispherical-directional reflectance factor (HDRF), both of which include radiation from the entire hemisphere and not just from a single

direction. The hemispherical-directional reflectance factor is formalized by:

$$
\begin{aligned}
R_{\mathrm{HDRF}} &= \frac{\mathrm{d}I_{\mathrm{r}}^{\uparrow}(\theta_{\mathrm{i}}, \varphi_{\mathrm{i}}, 2\pi; \theta_{\mathrm{r}}, \varphi_{\mathrm{r}})}{\mathrm{d}F_{0}^{\downarrow}(\theta_{\mathrm{i}}, \varphi_{\mathrm{i}}, 2\pi)} \\
&= R_{\mathrm{HDRF}}(\theta_{\mathrm{i}}, \varphi_{\mathrm{i}}; \theta_{\mathrm{r}}, \varphi_{\mathrm{r}}) \cdot f_{\mathrm{dir}} \\
&\quad + R(2\pi; \theta_{\mathrm{r}}, \varphi_{\mathrm{r}}) \cdot (1 - f_{\mathrm{dir}}).
\end{aligned}
\tag{B2}
$$

Thus, $R_{\mathrm{HDRF}}$ represents the ratio of the reflected incident radiation from a surface in relation to an ideal, lambertian surface. Furthermore, in Eq. B2 it is assumed that the incident, diffuse radiation is isotropic (Schaepman-Strub et al., 2006).

Figure B1a and e show $R_{\mathrm{HDRF}}$ at 550 nm wavelength for cloud-free conditions with $\tau = 0$ for $\theta$ of 25° and 70°, respectively.

Scattering by gas molecules and aerosol particles lead to direct fraction $f_{\mathrm{dir}}(\lambda) = 0.94$ just below 1. Under these conditions the first term in Eq. B2 dominates and the directional effect of the incoming radiation on the reflected radiation is most pronounced leading to the clear development of the hot spot that follows the solar zenith angle of the Sun (see white markers). With increasing values of $\tau$ and decreasing $f_{\mathrm{dir}}(\lambda)$, the second term in Eq. B2 dominates since the radiation reaching the surface is predominantly diffuse and almost isotropic. The directional component of the surface reflectance vanishes, causing the hot

spot to disappear and leading to a general smoothing of $R_{\mathrm{HDRF}}$, which is also confirmed in the cross-sections in panels c, d, g, and h.

*Author contributions.* **KW** designed and implemented the model coupling, performed the simulations, and drafted the manuscript. **EJ**, **AE**, and **MW**, contributed to the preparation and revisions of the manuscript. **MS**, **HF**, and **AHu** supported during the model set-up and provided suggestions for the manuscript.

*Competing interests.* The authors declare no competing interest.

*Acknowledgements.* We thank the German Centre for Integrative Biodiversity Research (iDiv) Halle-Jena-Leipzig, which is a research center of the Deutsche Forschungsgemeinschaft (DFG). We also thank the Saxon State Ministry for Science, Culture and Tourism (SMWK) for funding through grant 3-7304/44/4-2023/8846. We would also like to acknowledge the three anonymous reviewers whose comments helped to improve the final manuscript.

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
