# Peer review of "Biases in estimated vegetation indices from observations under cloudy conditions"

_EGUsphere, 2025_

## Author Comment (AC1)

**Reply to Reviewer #1**
(Referee comment on "Biases in estimated vegetation indices from observations under cloudy conditions" by K. Wolf et al. (egusphere-2025-2082),https://doi.org/10.5194/egusphere-2025-2082-RC1, 2025)
* * *
**Dear Dr. Wolf and co-authors,**

**Thank you for developing this framework to address the cloud influence on ground reflectance measurements. Indeed, in the remote sensing community, our field measurements are highly dependent on the illumination conditions, frequently altered by clouds. I have several minor remarks for your consideration.**

**In eq. 1, 4, 6 what do you mean by the 'sr' argument? Pi is already assumed to be in steradian (sr) units, cancelling steradian in the upwelling radiance I.**
While in some publications π is considered to have the unit "sr", we disagree in the point that this is always the case. In our case, π and it's unit result from integrating over all solid angles of the hemisphere and assuming an isotropic / Lambertian surface. Thus there is no "sr" related to π. Therefore, we kept the notation of "sr" in equations 1, 4, and 6.

**A bit on the same line, multiplication by pi suggests that the surface reflects homogeneously in all directions, Lambertian reflectance. How big would you expect the influence of the directionality of the actual surface to be on the reflectance value?**
The Reviewer is right that we have to be more precise with the definition of reflectivity. The paragraph to define the reflectivity has been update as follows:
*"All remotely sensed VIs rely on the spectral reflectivity ρ(λ). In its most general form, ρ(λ) is defined as the ratio of the surface-reflected radiance from within an infinite solid angle to the incoming radiance from within an infinite solid angle (Nicodemus, 1977; Martonchik et al., 2000; Schaepman-Strub et al., 2006). The amount of radiation reflected into a given solid angle is defined by the surface-specific bidirectional reflectance distribution function (BRDF, Schaepman-Strub et al., 2006). Furthermore, the surface reflected radiation depends on topography (Matsushita et al., 2007) and on changes in illumination conditions determined by solar zenith angle, aerosol particles, and clouds (Singh and Frazier, 2018). Definitions of ρ(λ) based on first principals are given by Nicodemus (1977) and Schaepman-Strub et al. (2006). For brevity and simplicity, we restrict the definition of ρ(λ), assuming idealized conditions of pure diffuse illumination and Lambertian reflection at the surface, which results in: [...]"*

Under natural conditions, incoming radiation is neither purely diffuse nor purely direct. In such cases, the deviation from a Lambertian surface is expressed by the hemispherical-directional reflectance function (HDRF). SCOPE2.0 internally considers for the HDRF, since we prescribe the direct and diffuse irradiance obtained from libRadtran, and specify the solar zenith and azimuth angles, as well as the observation zenith and azimuth angles. This allows to obtain radiances that would be observed under field conditions, demonstrating the advantage of coupling SCOPE2.0 with libRadtran.

To make this more clear, we added the following sentence to section 2.2.2 Vegetation radiative transfer model SCOPE2.0:
*"The angular dependence of $I^\uparrow(\lambda)$ is considered for by the actual illumination and observation geometries, the direct and diffuse $F^\downarrow(\lambda)$, and the internal calculation of the reflectivity in SCOPE2.0. "*

Figure A1 in the Appendix of the submitted manuscript shows the normalized hemispherical–directional reflectance factor $R_{HDRF}$ at 550 nm wavelength. The plot displays $R_{HDRF}$ for different illumination conditions, which are specified by the cloud optical thickness $\tau$ and two solar zenith angles $\theta_i$, and resulting direct to diffuse ratios $f_{dir}$.
The polar plots and particularly the cross-sections show that the actual surface reflectivity deviates strongly from the isotropic assumption. In the general case, the amount of radiation that is reflected in a specific direction depends on the angles of the incoming irradiance, as it is given in equations A1 and A2 in the Appendix. The dependence on viewing geometry is pronounced along the principal plane, which is defined along the 0°-180° azimuth line, with the Sun positioned at 0°. For the example given in Fig. A1 a-d with solar zenith angle $\theta_i$ of 25° and $\tau$=0, a change in the viewing direction from nadir ($\theta_r$=0°) towards the hot spot at 25°, observed reflectivity would almost double. With increasing values of $\tau$ the pronounced effect of the hot spot vanishes but the effects towards large viewing angles become more pronounced (see Fig. A1 d). Even greater difference in between nadir observation and more slant observation geometries appear with increasing values of $\theta_i$ (see Fig. A1 e-h). Also other azimuth directions off the principle plane show a strong dependence on the viewing geometry (see Fig A1 right most column)

**On the SCOPE model (section 2.2.2 and Table 2).**

- **First of all, it is unclear why SCOPE was chosen instead of SAIL or INFORM. The latter is more suitable for pine forest (L163) simulations, as it explicitly has the concept of trunks and branches in it. SCOPE has energy balance, thermal domain, photosynthesis and chlorophyll fluorescence that other models do not have. Using it as a tool for a single reflectance simulation is overkill. Nonetheless, your choice.**
  Thank you for providing your model suggestions. However, we are aware that several models for vegetation radiative transfer exist.
  To our knowledge, an equivalent model to SCOPE2.0 would be, for example, PROSAIL, which combines PROSPECT (for leaf optical properties) and SAIL (for radiative transfer in the canopy). We chose SCOPE2.0 over other models because it provides an accessible way

to couple it with libRadtran, without the need to make fundamental changes in the code base of both models. Another factor is the ability of SCOPE2.0 to provide simulations in the thermal wavelength range, a potential topic that we are interested in future investigations. Using SCOPE2.0, now for the visible and near-infrared, and later for the thermal infrared, allows us to use the same or similar model framework for both wavelength ranges with only minor modifications.

We also acknowledge that INFORM may be better suited for an erectophile leaf angle distribution, but since we also simulated spherical and planophile leaf angle distribution, SCOPE2.0 appears to be a reasonable choice to us.

- **Why was only half of the important SCOPE input parameters chosen? The BSM model, for example, has brightness, two shape parameters and moisture content, but only B is shown in Table 2. In any case, those parameters are set to their default values so it is also not clear why highlighting them at all.**
  In the companion paper by Wolf et al. (2025a) [https://bg.copernicus.org/articles/22/2909/2025/bg-22-2909-2025.html], we presented a sensitivity study of selected SCOPE parameters relevant to canopy optical properties within the visible and near-infrared wavelength range. In the sensitivity study, the selected variables have been varied around their default values and the effects on the surface reflectivity and canopy albedo have been determined. Based on the relevance of these factors, ordered by their magnitude, they were selected to be included in Table 2 in the current manuscript. While some of the parameter default values have been modified to represent forest, others were kept constant, like the BSM parameter. The default value of the BSM model is listed because it was found to be an influential parameter in surface reflectivity. To better clarify the selection of parameters and the values, we added the following sentences to section "2.2.2 Vegetation radiative transfer model SCOPE2.0".
  *"Table 2 provides an overview of the selected parameters for the vegetation RT simulations. The parameters were selected based on their relevance to surface reflectivity within the visible and near-infrared wavelength ranges. Their individual relevance was estimated in a sensitivity study by Wolf et al. (2025a)."*

- **Finally, could you please be more explicit about which SCOPE output was integrated with the libRadtran output and how? I am a bit confused because L152 says *"As an initial guess of the surface albedo in libRadtran, the "mixed-forest" albedo was taken from the IGBP data base."* According to my understanding, it was sufficient to take some spectral forest reflectance for the exercise instead of running an RTM.**
  The Reviewer is right that we were not very clear about the iterative nature of the model coupling.
  The introductory subsection "2.2 Radiative transfer simulations" was modified, now including a rephrased sentence, which reads as follows:
  *"Furthermore, radiation interactions may occur between the surface and the cloud, which can be accounted for by **iterative** coupling of the RT models of the atmosphere and vegetation (Wolf et al., 2025a). In the present paper we use the same model coupling setup*

*introduced and described by Wolf et al. (2025a)".*

To clarify the confusion about the initial guess of surface albdo:
When the atmosphere radiative transfer (RT) model libRadtran is run for the first time, it is initialized with a first guess for surface albedo, which is taken from the IGBP database. After that, SCOPE2.0 is run with the provided downward irradiance from libRadtran. In the second iteration, the initial guess is replaced by the surface albedo that is based on the upward irradiance provided by SCOPE2.0 and the downward irradiance provided by libRadtran. Two iterations were found to be sufficient for the simulated cases.
To emphasize that the IGBP is only used in the initial run and is later replaced by the albedo determined from the coupled atmosphere-vegetation model, the following sentence in section "2.2.1 Atmospheric radiative transfer model libRadtran", was modified:
*"The iteration process was first started by running libRadtran, with an initial guess for the surface albedo. The "mixed-forest" albedo was taken from the IGBP database (Loveland and Belward, 1997). After one iteration cycle, the surface albedo determined during the iterative model coupling process was used (Wolf et al., 2025a)."*
We hope that this answers the Reviewer's question. However, we do refrain from providing a more detailed description of the model coupling and implementation in the submitted manuscript because the coupling is described in depth in the companion paper by Wolf et al 2025.

**Figure 2b. Please, add a legend.**
Figure 2a and b share the same legend. To clarify this, the following sentence has been added to the figure caption.
*"Panels (a) and (b) share the same legend."*

**L187 – "Wolf et al. (2024) have shown the influence of clouds on direct and diffuse $F{\downarrow}(\lambda)$, the associated effects on $F{\uparrow}(\lambda)$," please, write exactly what the influence was. I guess more clouds – more diffuse radiation.**
Yes, this is correct, Wolf et al. (2024) showed, that clouds increase the diffuse radiation. An increase in cloud optical thickness $\tau$ leads to a decrease in direct irradiance $F^{\downarrow}_{dir}(\lambda)$. However, the response of the diffuse irradiance $F^{\downarrow}_{dif}(\lambda)$ depends on $\tau$. For values of $\tau$ below 4 to 6, the diffuse irradiance first increases, and then decreases as $\tau$ increases further. The total amount of $F^{\downarrow}(\lambda)$, direct plus diffuse $F^{\downarrow}(\lambda)$, decreases as $\tau$ increases, with an increasing fraction of diffuse radiation.
Furthermore, an increase in the diffuse fraction reduces the influence of changes in the solar zenith angle on the upward irradiance. Lastly, the presences of clouds shifts the weighting of the incoming radiation towards shorter wavelengths, as clouds primarily absorb radiation at longer wavelengths.
The paragraph in section 3.1 has been rephrased as follows:
*"Wolf et al. (2025a) have shown the influence of clouds on direct and diffuse $F^{\downarrow}(\lambda)$, the effects on $F^{\uparrow}(\lambda)$, and the resulting albedo effects over vegetated areas using coupled atmosphere–vegetation radiative transfer models. An increase in $\tau$ leads to a decrease in $F^{\downarrow}dir(\lambda)$, while the response of $F^{\downarrow}dif(\lambda)$ depends on $\tau$. For values of $\tau$ less than 4 to 6, $F^{\downarrow}dif(\lambda)$ first increases and then decreases as $\tau$ increases further. The total $F^{\downarrow}(\lambda)$ and $f_{dir}(\lambda)$ both continuously decrease as $\tau$ increases. In addition, $F^{\uparrow}(\lambda)$ became less sensitive to changes in $\theta$. Lastly, the presence of clouds modulates the*

*incoming radiation spectrally by shifting the incoming radiation towards shorter wavelengths, as clouds primarily scatter radiation at shorter wavelength and absorb radiation at longer wavelengths. Wolf et al. (2025a) also showed that radiative interactions between the canopy and the cloud base increase $F^{\downarrow}dif\,(\lambda)$ and albedo compared to cloud-free conditions. The present paper focuses on the related effects on $I^{\uparrow}(\lambda)$ and $\rho(\lambda)$."*

**Figure 3. What do grey areas show? Sentinel-2 bands?**

*"The gray marked areas highlight the Sentinel-2 bands B2, B4, B8, B8a, and B11."* The very same sentence was added to the figure captions of Fig. 2 and Fig. 3.

**Figures 3 and 4 captions. Please, note, you are working with synthetic (modelled) data. Remove the term "measured" reflectance; do not mislead the readers.**

To be more clear, we added the word "synthetic" to all instances, where we refereed to "measurements" to make clear that we refer to the simulated measurements. The sentence in caption of Fig 3 was changed to: *"... and constant cloud optical thickness during calibration and the actual synthetic measurements ..."*

**Figure 5. Please, check the location of symbols inside the heatmaps Whereas for NDVI (circle) lambda1 and lambda2 are matching the expected NIR and RED, NDWI1240 is definitely far from lambda1=1240 nm. Furthermore, the symbol in Figures 5c and 5d around lambda1=900nm, lambda2=1600nm is unclear (or absent from the legend).**

Thank you for pointing this out. The second Reviewer had a similar comment. The figure has been revised, now with the correct position of the markers and the completed legend. The color-style has been adjusted to improve the legibility of the markers.

---

## Author Comment (AC2)

**Reply to Reviewer #2**
(Referee comment on "Biases in estimated vegetation indices from observations under cloudy conditions" by K. Wolf et al. (egusphere-2025-2082),https://doi.org/10.5194/egusphere-2025-2082-RC2, 2025)

We would like to thank the Reviewer for taking the time to review the manuscript and for providing comments that helped us improve it. Below, we respond to the Reviewer's comments. For clarity, the Reviewer's comments are in **bold** and the changes to the manuscript are *in italics*. Please note that some additional changes have also been made to improve the writing and style.
* * *
**This manuscript reports on the effect of clouds on the derivation of vegetation indices from remote sensing platforms in a radiative transfer modelling approach. Basically it is a sensitivity study on how clouds contribute to the incoming radiation and what effect it might have on the calibration conditions for deriving the reflectance values and consequently on the computation of the various VI with Sentinel 2 bands as example for different solar zenith angles.**

**Overall, it is an interesting and relevant study since it brings our attention to the basics of using reflectance values and its impact on derived vegetation indices. It is important that we do not forget the basics of RS data. The preprint is very well written and well-documented, with nice figures.**

Thank you for the generally positive feedback.

**To my opinion, the authors should elaborate a bit more on the consequences of the effect on biases in values of VI's. They briefly touch the impact on LAI derived from EVI, but the authors should add some paragraph how it could impact remotely sensed derived biophysical properties such as GPP, green water fluxes etc with proper referencing. This would make the paper even more relevant. Perhaps, indicating some of the effects as percentages can increase the visibility.**
We added a subsection titled "3.4 Implication of biases in vegetation indices on estimated biophysical properties" in accordance with the Reviewer's suggestion. The added subsection discusses the impact of biases in the enhanced vegetation index (EVI) and the normalized differential vegetation index (NDVI) on the estimation of gross primary production, fresh and dry biomass, vegetation water content, and leaf area index. We added relevant citations in which the correlations between EVI, NDVI, and the biophysical properties are derived. Due to the length of the subsection, we would like to direct the Reviewer to the track changes file.

**In lines 117, 306 and others, the authors directly link NDVI to vegetation health. I would refrain from that, since NDVI essentially says something about the "greenness" of the**

**vegetation, but not necessarily on its health. It can be used for vegetation health, but it is not synonymous for health.**

We considered the Reviewer's comment and rewrote all instances where the NDVI was directly linked with "vegetation health". The sentences were rephrased as follows:

*"The greatest variability is found for θ = 25°, where NDVI increases from 0.87 to 0.91 with increasing τ , which could be interpreted as an overestimation of vegetation health."*

*"For $τ_{cal}$ = 0 an extreme increase of $τ_{mea}$ from 0 to 40 results in a decrease in NDVI from 0.87 to 0.84, which could be interpreted as an underestimation of vegetation health"*

*"In both cases, the inferred ground-truth vegetation health would be overestimated."*

**Furthered, I only have few minor/textual comments**

**Normally, numbers less than 10 are written as text; So less than one; equals one, etc**

We followed the suggestion of the Reviewer and scanned the text for these mistakes. However, during previous submissions to Copernicus journals, phrases like "...ranges from 0 to 1..." were accepted by copy-editing and typesetting. Therefore we kept this stile of writing. Similarly, we ket instance such as "...smaller than 1", since those were also accepted by Copernicus copy-editing and type-setting.  The Copernicus guidelines say: "For items other than units of time or measure, use words for cardinal numbers less than 10; use numerals for 10 and above (e.g. three flasks, seven trees, 6 m, 9 d, 10 desks)."

**If one uses for instance 0.29, also use 0.20 and not 0.2 (see L338, but also elsewhere): always use the same amount of decimals for the same property**

We acknowledge this comment. However, we did not find a specific guideline that says that all values have to use the same number of digits. The SI-guidelines say that trailing zeros should be used when one wants to imply a certain precision of, e.g., a measuring device. During previous submissions to Copernicus journals, trailing zeros have been removed. Since there are no clear rules, we would like to leaf it to typesetting and copy-editing of the journal.

**I do not know the policy of the journal, but normally the cited references in the text are first ordered chronologically and then alphabetically**

We agree with the Reviewer that citations should be in chronological order. This was an error on our part, and we have corrected the order of the citations.

**Abstract: add more on possible consequences;**

In line with the Reviewer's first comment, we added the following sentence to the abstract:

*"Other estimates of biophysical properties derived from EVI, such as gross primary product, fresh and dry biomass, or vegetation water content, are similarly affected."*

**Captions Fig 1. Replace "'relate" with "connect"**

We followed the Reviewer's suggestion.

**L30: specify what tau is under the text of Eq 1; see lines 45-47; this should come earlier in the text**

The Reviewer is right and the definition of the cloud optical thickness is given earlier in the text.
*"The cloud optical thickness τ(λ) is a measure of the extinction of radiation for a vertical path through the cloud, serving as vertical coordinate."*

**L61: "attempts"? This is not a proper use of the word here;**

The word "attempts" has been replaced with "allows." The sentence now reads:
*"Using a RP enables transfer calibration."*

**L67: In general, it is a "transfer function" rather than a factor, although used as a factor;**

To be more precise, we followed the suggestion of the Reviewer and replaced "transfer factor" with "transfer function".

**L76: Should be "requires frequent calibrations of the transfer function";**

We partially adopted the Reviewer's suggestion and modified the sentence as follows: "...requires frequent RP overflights to obtain updated transfer functions."

**L217: remove "the" before Appendix A;**

"The" has been removed.

**L224: "valueS";**

The typo has been corrected.

**L270-271: "ARE close to zero"**

The sentence has been corrected.

**Figure 5: Why is the symbol of NDII missing in the upper right panel?**

The second Reviewer had a similar comment in this regard. Figure 5 has now been revised with the markers in the correct position and a completed legend, and the intensity of the color scale has been adjusted to enhance the legibility of the markers.

**L281, 324 and other lines: I would refrain of using the term "exemplary"; it echoes a bit as "exemplary behavior or punishment"; Perhaps use "illustrates"?**

Based on the Reviewer's suggestion, all instances of "exemplary" have been rephrased.

**L345: Should be "The effect of changes in fdir";**

The text has been modified as suggested.

**L378-379: "… was between 0 and 40," and ".. ranged …"; Remove "were covered":**

The text has been modified as suggested.

**L406: remove comma after NDWI**

As suggested, the comma was removed.

**L407: twice increase, increasing;**

The second instance of "twice" was removed.

---

## Editor Decision (ED1)

**GENERAL COMMENTS**

This study makes use of a coupled atmospheric-vegetation radiative transfer model to simulate the effects of (homogeneous) clouds on three vegetation índices computed from drone imagery. To do so, the authors simulate a single canopy scenario with the optical radiative transfer model of SCOPE, and different illumination conditions (based on different combinations of sun zenith angle and cloud optical thickness). They present how the reflectance factors and the derived spectral índices are affected by these conditions and by changes in illumination between the measurement of reference panels and targets.

Despite the potential complexity of coupling two RTMs, which is part of another work under review, I fear I do not see any scientific added value in this work. I have the impresión that the results presented can be directly inferred from already known theory, and that the problem stated for the "drone" world has been dealt with for a long time in the area of field spectroradiometry. In fact, many of the statements referenced as findings from a former manuscript by the author (Wolf et al., 2024) are well-established in atmospheric radiative transfer and BRDF theory (e.g., "Therefore, $\rho(\lambda)$ is also sensitive to $\theta$ (see Wolf et al., 2024), Appendix D)" is already known in BRDF theory; also, the comments on the dependence of albedo on the fraction of direct/diffuse incoming irradiance are described, for example, by Pinty et al., 2005). Any novelty brought by this manuscript does not stand out from older theoretical knowledge, as presented.

At the same time, the range of simulated scenarios is so narrow (i.e., only one canopy) that the results presented cannot be generalized or used to provide any recommendations. In fact, the results could be different for canopies or surfaces of varying properties (e.g., see the effect of a bowl- or bell-shaped BRDF in Pinty et al., 2005). Since no generalizable knowledge can be obtained, the "cautions" arising from this work offer no added value to the knowledge already generated by the proximal sensing community. Additionally, the authors do not provide a drone-specific correction method, which, on the other hand, would already be known from field spectroscopy: continuous monitoring of irradiance and BRDF characterization, or just avoiding unstable conditions. For some of the scenarios described, proximal sensing would just discard the measurements due to changes in illumination. Maybe drone operators should do the same. Moreover, drone-specific issues, such as the distance between the target and the sensor, where this modeling could offer something new (although this is also a long-known issue for airborne imagers), are often overlooked.

While the authors acknowledge the limitations of the simulations, the justification provided - "Nevertheless, we argue that the presented study can be used as a first approximation for the transition between cloud-free and cloudy regions"- is insufficient a scientific publication; particularly when no new knowledge is generated, and the claimed sunny to cloudy transition (characterized by heterogeneous cloud cover as these develop or arrive from the horizon) is not represented by the simulation of homogeneous clouds they perform.

I have the feeling that the authors have invested time in building a tool (Wolf et al., 2024), but they have not yet found a way to make a significant contribution with it, at least in this case. I recommend stepping back, reviewing former works on this topic, and identifying a problem that this tool can genuinely contribute to solving or alleviating.

**References.**

* Pinty, B., Lattanzio, A., Martonchik, J. V., Verstraete, M. M., Gobron, N., Taberner, M., Widlowski, J.-L., Dickinson, R. E., and Govaerts, Y.: Coupling Diffuse Sky Radiation and Surface Albedo, Journal of the Atmospheric Sciences, 62, 2580-2591, https://doi.org/10.1175/JAS3479.1, 2005.

* Wolf, K., Jäkel, E., Ehrlich, A., Schäfer, M., Feilhauer, H., Huth, A., Weigelt, A., and Wendisch, M.: Impact of clouds on vegetation albedo quantified by coupling an atmosphere and a vegetation radiative transfer model, EGUsphere, 2024, 1–30, https://doi.org/10.5194/egusphere-2024-3614, 2024.

**SPECIFIC COMMENTS**

**Reflectivity:** The term reflectivity is misused (it is rather an intrinsic property of the matter, whereas the reflectance and related terms are sample-specific), and for sure, SCOPE does not simulate reflectivity. The right term to be used in this work is "reflectance factor" (see Schaepman-Strub et al., 2006 for a definition).

* Schaepman-Strub, G., Schaepman, M. E., Painter, T. H., Dangel, S., and Martonchik, J. V.: Reflectance quantities in optical remote sensing—definitions and case studies, Remote Sensing of Environment, 103, 27-42, https://doi.org/10.1016/j.rse.2006.03.002, 2006.

**Equation 1**: The term "sr" is not defined, "$\tau$" is not defined in the subsequent paragraphs. Overall, make sure all terms are defined the first time they are introduced in the text.

**Lines 39-40:** What reference supports this statement? Reflectance factors can be larger than one, but reflectance (the ratio of the total reflected to total incoming radiance) cannot (see again Schaepman-Strub et al., 2006). If so, are "broken clouds" what make reflectance factors go beyond one? If that has been proven, it must be referenced (and also other causes).

**Lines 41:** Remote sensing does not retrieve vegetation índices. It retrieves reflectance factors, from which vegetation índices can be computed.

**Equation 4:** Misses the scaling factor of the integration time or gains set for the sensor and ignores the dark current that must be subtracted from the recorded digital signal.

**Lines 69-74**: This explanation is nuclear. There are several ways to prepare for a drone flight. Sometimes, the operator holds the camera above a reference panel (by the way, usually grey, and the darker the less Lambertian, assuming in the best case situation Spectralon® is used) to optimize the integration time and get a cross-calibration between the up-welling and the down-welling channels of the sensing sistema if the second exists, which is not always the case. The alternative is to place large reference panels in the study area and fly above them in one of the overpasses, using them as a reference. It is not clear what procedure the authors describe.

**Lines 153-154:** The fact that libRadtran simulates down-welling irradiance at 40 m above ground means that no differences between the irradiance level at the drone and at the target height are considered. This should be clarified, justified, and the potential impact on the results discussed.

**Lines 212-213:** "While direct radiation is reflected into a narrow solid angle, diffuse radiation is scattered over a wider solid angle (Schaepman-Strub et al., 2006)." This statement is false. Direct radiation is reflected in all directions (e.g., a Lambertian diffuser). A different thing is that a Bi-directional reflectance factor is defined as the ratio between the incoming and outgoing radiance in infinitesimal solid angles.

---

## Author Response (AR2)

**Reply to Reviewer #1**
(Referee comment on "Biases in estimated vegetation indices from observations under cloudy conditions" by K. Wolf et al. (egusphere-2025-2082),https://doi.org/10.5194/egusphere-2025-2082-RC1, 2025)

We would like to thank the Reviewer for taking the time to review the manuscript and for providing comments that helped us improve it. Below, we respond to the Reviewer's comments. For clarity, the Reviewer's comments are in **bold** and the changes to the manuscript are *in italics*. Please note that some additional changes have also been made to improve the writing and style.
* * *
**Dear Dr. Wolf and co-authors,**

**Thank you for developing this framework to address the cloud influence on ground reflectance measurements. Indeed, in the remote sensing community, our field measurements are highly dependent on the illumination conditions, frequently altered by clouds. I have several minor remarks for your consideration.**

**In eq. 1, 4, 6 what do you mean by the 'sr' argument? Pi is already assumed to be in steradian (sr) units, cancelling steradian in the upwelling radiance I.**
While in some publications π is considered to have the unit "sr", we disagree in the point that this is always the case. In our case, π and it's unit result from integrating over all solid angles of the hemisphere and assuming an isotropic / Lambertian surface. Thus there is no "sr" related to π. Therefore, we kept the notation of "sr" in equations 1, 4, and 6.

**A bit on the same line, multiplication by pi suggests that the surface reflects homogeneously in all directions, Lambertian reflectance. How big would you expect the influence of the directionality of the actual surface to be on the reflectance value?**
The Reviewer is right that we have to be more precise with the definition of reflectivity, now termed 'reflectance' to account for Reviewer 3. The paragraph to define the reflectance has been update as follows:
*"All remotely sensed VIs rely on the spectral reflectance ρ(λ). In its most general form, ρ(λ) is defined as the ratio of the surface-reflected radiance from within an infinite solid angle to the incoming radiance from within an infinite solid angle (Nicodemus, 1977; Martonchik et al., 2000; Schaepman-Strub et al., 2006). The amount of radiation reflected into a given solid angle is defined by the surface-specific bidirectional reflectance distribution function (BRDF, Schaepman-Strub et al., 2006). Furthermore, the surface reflected radiation depends on topography (Matsushita et al., 2007) and on changes in illumination conditions determined by solar zenith angle, aerosol particles, and clouds (Pinty et al., 2005; Singh and Frazier, 2018). Definitions of ρ(λ) based on first principals are given by Nicodemus (1977) and Schaepman-Strub et al. (2006). For brevity and simplicity, we restrict the definition of ρ(λ), assuming idealized conditions of pure diffuse*

*illumination and Lambertian reflection at the surface, which results in: [...]"*

Under natural conditions, incoming radiation is neither purely diffuse nor purely direct. In such cases, the deviation from a Lambertian surface is expressed by the hemispherical-directional reflectance function (HDRF). SCOPE2.0 internally considers for the HDRF, since we prescribe the direct and diffuse irradiance obtained from libRadtran, and specify the solar zenith and azimuth angles, as well as the observation zenith and azimuth angles. This allows to obtain radiances that would be observed under field conditions, demonstrating the advantage of coupling SCOPE2.0 with libRadtran.

To make this more clear, we added the following sentence to section 2.2.2 Vegetation radiative transfer model SCOPE2.0:
*"The angular dependence of $I^\uparrow(\lambda)$ is considered for by the actual illumination and observation geometries, the direct and diffuse $F^\downarrow(\lambda)$, and the parameterized intrinsic reflectivity within SCOPE2.0. "*

Figure A1 in the Appendix of the submitted manuscript shows the normalized hemispherical–directional reflectance factor $R_{HDRF}$ at 550 nm wavelength. The plot displays $R_{HDRF}$ for different illumination conditions, which are specified by the cloud optical thickness $\tau$ and two solar zenith angles $\theta_i$ , and resulting direct to diffuse ratios $f_{dir}$.
The polar plots and particularly the cross-sections show that the actual surface reflectance deviates strongly from the isotropic assumption. In the general case, the amount of radiation that is reflected in a specific direction depends on the angles of the incoming irradiance,  as it is given in equations A1 and A2 in the Appendix. The dependence on viewing geometry is pronounced along the principal plane, which is defined along the 0°-180° azimuth line, with the Sun positioned at 0°. For the example given in Fig. A1 a-d with solar zenith angle $\theta_i$ of 25° and $\tau$=0, a change in the viewing direction from nadir ($\theta_r$=0°) towards the hot spot at 25°, observed reflectance would almost double. With increasing values of $\tau$ the pronounced effect of the hot spot vanishes but the effects towards large viewing angles become more pronounced (see Fig. A1 d). Even greater difference in between nadir observation and more slant observation geometries appear with increasing values of $\theta_i$ (see Fig. A1 e-h). Also other azimuth directions off the principle plane show a strong dependence on the viewing geometry (see Fig A1 right most column)

**On the SCOPE model (section 2.2.2 and Table 2).**

- **First of all, it is unclear why SCOPE was chosen instead of SAIL or INFORM. The latter is more suitable for pine forest (L163) simulations, as it explicitly has the concept of trunks and branches in it. SCOPE has energy balance, thermal domain, photosynthesis and chlorophyll fluorescence that other models do not have. Using it as a tool for a single reflectance simulation is overkill. Nonetheless, your choice.**
  Thank you for providing your model suggestions. However, we are aware that several models for vegetation radiative transfer exist.
  To our knowledge, an equivalent model to SCOPE2.0 would be, for example, PROSAIL, which combines PROSPECT (for leaf optical properties) and SAIL (for radiative transfer in

the canopy). We chose SCOPE2.0 over other models because it provides an accessible way to couple it with libRadtran, without the need to make fundamental changes in the code base of both models. Another factor is the ability of SCOPE2.0 to provide simulations in the thermal wavelength range, a potential topic that we are interested in future investigations. Using SCOPE2.0, now for the visible and near-infrared, and later for the thermal infrared, allows us to use the same or similar model framework for both wavelength ranges with only minor modifications.

We also acknowledge that INFORM may be better suited for an erectophile leaf angle distribution, but since we also simulated spherical and planophile leaf angle distribution, SCOPE2.0 appears to be a reasonable choice to us.

- **Why was only half of the important SCOPE input parameters chosen? The BSM model, for example, has brightness, two shape parameters and moisture content, but only B is shown in Table 2. In any case, those parameters are set to their default values so it is also not clear why highlighting them at all.**
  In the companion paper by Wolf et al. (2025a) [https://bg.copernicus.org/articles/22/2909/2025/bg-22-2909-2025.html], we presented a sensitivity study of selected SCOPE parameters relevant to canopy optical properties within the visible and near-infrared wavelength range. In the sensitivity study, the selected variables have been varied around their default values and the effects on the surface reflectance and canopy albedo have been determined. Based on the relevance of these factors, ordered by their magnitude, they were selected to be included in Table 2 in the current manuscript. While some of the parameter default values have been modified to represent forest, others were kept constant, like the BSM parameter. The default value of the BSM model is listed because it was found to be an influential parameter in surface reflectance. To better clarify the selection of parameters and the values, we added the following sentences to section "2.2.2 Vegetation radiative transfer model SCOPE2.0".
  *"Table 2 provides an overview of the selected parameters for the vegetation RT simulations. The parameters were selected based on their relevance to surface reflectance within the visible and near-infrared wavelength ranges. Their individual relevance was estimated in a sensitivity study by Wolf et al. (2025a)."*

- **Finally, could you please be more explicit about which SCOPE output was integrated with the libRadtran output and how? I am a bit confused because L152 says *"As an initial guess of the surface albedo in libRadtran, the "mixed-forest" albedo was taken from the IGBP data base."* According to my understanding, it was sufficient to take some spectral forest reflectance for the exercise instead of running an RTM.**
  The Reviewer is right that we were not very clear about the iterative nature of the model coupling.
  The introductory subsection "2.2 Radiative transfer simulations" was modified, now including a rephrased sentence, which reads as follows:
  *"Furthermore, radiation interactions may occur between the surface and the cloud, which can be accounted for by **iterative** coupling of the RT models of the atmosphere and*

*vegetation (Wolf et al., 2025a). In the present paper we use the same model coupling setup introduced and described by Wolf et al. (2025a)".*

To clarify the confusion about the initial guess of surface albdo:
When the atmosphere radiative transfer (RT) model libRadtran is run for the first time, it is initialized with a first guess for surface albedo, which is taken from the IGBP database. After that, SCOPE2.0 is run with the provided downward irradiance from libRadtran. In the second iteration, the initial guess is replaced by the surface albedo that is based on the upward irradiance provided by SCOPE2.0 and the downward irradiance provided by libRadtran. Two iterations were found to be sufficient for the simulated cases.
To emphasize that the IGBP is only used in the initial run and is later replaced by the albedo determined from the coupled atmosphere-vegetation model, the following sentence in section "2.2.1 Atmospheric radiative transfer model libRadtran", was modified:
*"The iteration process was first started by running libRadtran, with an initial guess for the surface albedo. The "mixed-forest" albedo was taken from the IGBP database (Loveland and Belward, 1997). After one iteration cycle, the surface albedo determined during the iterative model coupling process was used (Wolf et al., 2025a)."*
We hope that this answers the Reviewer's question. However, we do refrain from providing a more detailed description of the model coupling and implementation in the submitted manuscript because the coupling is described in depth in the companion paper by Wolf et al 2025.

**Figure 2b. Please, add a legend.**
Figure 2a and b share the same legend. To clarify this, the following sentence has been added to the figure caption.
*"Panels (a) and (b) share the same legend."*

**L187 – *"Wolf et al. (2024) have shown the influence of clouds on direct and diffuse $F\downarrow(\lambda)$, the associated effects on $F\uparrow(\lambda)$,"* please, write exactly what the influence was. I guess more clouds – more diffuse radiation.**
Yes, this is correct, Wolf et al. (2024) showed, that clouds increase the diffuse radiation. An increase in cloud optical thickness $\tau$ leads to a decrease in direct irradiance $F^{\downarrow}_{dir}(\lambda)$. However, the response of the diffuse irradiance $F^{\downarrow}_{dif}(\lambda)$ depends on $\tau$. For values of $\tau$ below 4 to 6, the diffuse irradiance first increases, and then decreases as $\tau$ increases further. The total amount of $F^{\downarrow}(\lambda)$, direct plus diffuse $F^{\downarrow}(\lambda)$, decreases as $\tau$ increases, with an increasing fraction of diffuse radiation.
Furthermore, an increase in the diffuse fraction reduces the influence of changes in the solar zenith angle on the upward irradiance. Lastly, the presences of clouds shifts the weighting of the incoming radiation towards shorter wavelengths, as clouds primarily absorb radiation at longer wavelengths.
The paragraph in section 3.1 has been rephrased as follows:
*"Pinty et al. (2005) have shown the influence of molecules and aerosols on direct and diffuse $F^{\downarrow}(\lambda)$, the effects on $F^{\uparrow}(\lambda)$, and the resulting albedo effects over vegetated areas based on analytical equations. Wolf et al. (2025a) used coupled atmosphere–vegetation radiative transfer models to investigate these effects, but explicitly considering clouds and spectral depending scattering and absorption in the atmosphere. An increase in $\tau$ leads to a decrease in $F^{\downarrow}dir(\lambda)$, while the response of*

*$F^{\downarrow}_{dif}$ (λ) depends on τ . For values of τ less than 4 to 6, $F^{\downarrow}_{dif}$ (λ) first increases and then decreases as τ increases further. The total $F^{\downarrow}$(λ) and $f_{dir}$(λ) both continuously decrease as τ increases. In addition, $F^{\uparrow}$(λ) became less sensitive to changes in θ. Lastly, the presence of clouds modulates the incoming radiation spectrally by shifting the incoming radiation towards shorter wavelengths, as clouds primarily scatter radiation at shorter wavelength and absorb radiation at longer wavelengths. Wolf et al. (2025a) also showed that radiative interactions between the canopy and the cloud base increase $F^{\downarrow}_{dif}$ (λ) and albedo compared to cloud-free conditions. The present paper focuses on the related effects on $I^{\uparrow}$(λ) and ρ(λ)."*

**Figure 3. What do grey areas show? Sentinel-2 bands?**
*"The gray marked areas highlight the Sentinel-2 bands B2, B4, B8, B8a, and B11."* The very same sentence was added to the figure captions of Fig. 2 and Fig. 3.

**Figures 3 and 4 captions. Please, note, you are working with synthetic (modelled) data. Remove the term "measured" reflectance; do not mislead the readers.**
To be more clear, we added the word "synthetic" to all instances, where we refereed to "measurements" to make clear that we refer to the simulated measurements. The sentence in caption of Fig 3 was changed to: *"... and constant cloud optical thickness during calibration and the actual synthetic measurements ..."*

Figure 5. Please, check the location of symbols inside the heatmaps Whereas for NDVI (circle) lambda1 and lambda2 are matching the expected NIR and RED, NDWI1240 is definitely far from lambda1=1240 nm. Furthermore, the symbol in Figures 5c and 5d around lambda1=900nm, lambda2=1600nm is unclear (or absent from the legend).
Thank you for pointing this out. The second Reviewer had a similar comment. The figure has been revised, now with the correct position of the markers and the completed legend. The color-style has been adjusted to improve the legibility of the markers.

**Reply to Reviewer #2**
(Referee comment on "Biases in estimated vegetation indices from observations under cloudy conditions" by K. Wolf et al. (egusphere-2025-2082),[https://doi.org/10.5194/egusphere-2025-2082-RC2](https://doi.org/10.5194/egusphere-2025-2082-RC2), 2025)

We would like to thank the Reviewer for taking the time to review the manuscript and for providing comments that helped us improve it. Below, we respond to the Reviewer's comments. For clarity, the Reviewer's comments are in **bold** and the changes to the manuscript are *in italics*. Please note that some additional changes have also been made to improve the writing and style.
* * *
**This manuscript reports on the effect of clouds on the derivation of vegetation indices from remote sensing platforms in a radiative transfer modelling approach. Basically it is a sensitivity study on how clouds contribute to the incoming radiation and what effect it might have on the calibration conditions for deriving the reflectance values and consequently on the computation of the various VI with Sentinel 2 bands as example for different solar zenith angles.**

**Overall, it is an interesting and relevant study since it brings our attention to the basics of using reflectance values and its impact on derived vegetation indices. It is important that we do not forget the basics of RS data. The preprint is very well written and well-documented, with nice figures.**

Thank you for the generally positive feedback.

**To my opinion, the authors should elaborate a bit more on the consequences of the effect on biases in values of VI's. They briefly touch the impact on LAI derived from EVI, but the authors should add some paragraph how it could impact remotely sensed derived biophysical properties such as GPP, green water fluxes etc with proper referencing. This would make the paper even more relevant. Perhaps, indicating some of the effects as percentages can increase the visibility.**
We added a subsection titled "3.4 Implication of biases in vegetation indices on estimated biophysical properties" in accordance with the Reviewer's suggestion. The added subsection discusses the impact of biases in the enhanced vegetation index (EVI) and the normalized differential vegetation index (NDVI) on the estimation of gross primary production, fresh and dry biomass, vegetation water content, and leaf area index. We added relevant citations in which the correlations between EVI, NDVI, and the biophysical properties are derived. Due to the length of the subsection, we would like to direct the Reviewer to the track changes file.

**In lines 117, 306 and others, the authors directly link NDVI to vegetation health. I would refrain from that, since NDVI essentially says something about the "greenness" of the**

**vegetation, but not necessarily on its health. It can be used for vegetation health, but it is not synonymous for health.**

We considered the Reviewer's comment and rewrote all instances where the NDVI was directly linked with "vegetation health". The sentences were rephrased as follows:

*"The greatest variability is found for $\theta = 25°$, where NDVI increases from 0.87 to 0.91 with increasing $\tau$, which could be interpreted as an overestimation of vegetation health."*

*"For $\tau_{cal} = 0$ an extreme increase of $\tau_{mea}$ from 0 to 40 results in a decrease in NDVI from 0.87 to 0.84, which could be interpreted as an underestimation of vegetation health"*

*"In both cases, the inferred ground-truth vegetation health would be overestimated."*

**Furthered, I only have few minor/textual comments**

**Normally, numbers less than 10 are written as text; So less than one; equals one, etc**

We followed the suggestion of the Reviewer and scanned the text for these mistakes. However, during previous submissions to Copernicus journals, phrases like "...ranges from 0 to 1..." were accepted by copy-editing and typesetting. Therefore we kept this stile of writing. Similarly, we ket instance such as "...smaller than 1", since those were also accepted by Copernicus copy-editing and type-setting.  The Copernicus guidelines say: "For items other than units of time or measure, use words for cardinal numbers less than 10; use numerals for 10 and above (e.g. three flasks, seven trees, 6 m, 9 d, 10 desks)."

**If one uses for instance 0.29, also use 0.20 and not 0.2 (see L338, but also elsewhere): always use the same amount of decimals for the same property**

We acknowledge this comment. However, we did not find a specific guideline that says that all values have to use the same number of digits. The SI-guidelines say that trailing zeros should be used when one wants to imply a certain precision of, e.g., a measuring device. During previous submissions to Copernicus journals, trailing zeros have been removed. Since there are no clear rules, we would like to leaf it to typesetting and copy-editing of the journal.

**I do not know the policy of the journal, but normally the cited references in the text are first ordered chronologically and then alphabetically**

We agree with the Reviewer that citations should be in chronological order. This was an error on our part, and we have corrected the order of the citations.

**Abstract: add more on possible consequences;**

In line with the Reviewer's first comment, we added the following sentence to the abstract:

*"Other estimates of biophysical properties derived from EVI, such as gross primary product, fresh and dry biomass, or vegetation water content, are similarly affected."*

**Captions Fig 1. Replace "'relate" with "connect"**
We followed the Reviewer's suggestion.

**L30: specify what tau is under the text of Eq 1; see lines 45-47; this should come earlier in the text**
The Reviewer is right and the definition of the cloud optical thickness is given earlier in the text.
*"The cloud optical thickness τ(λ) is a measure of the extinction of radiation for a vertical path through the cloud, serving as vertical coordinate."*

**L61: "attempts"? This is not a proper use of the word here;**
The word "attempts" has been replaced with "allows." The sentence now reads:
*"Using a RP enables transfer calibration."*

**L67: In general, it is a "transfer function" rather than a factor, although used as a factor;**
To be more precise, we followed the suggestion of the Reviewer and replaced "transfer factor" with "transfer function".

**L76: Should be "requires frequent calibrations of the transfer function";**
We partially adopted the Reviewer's suggestion and modified the sentence as follows: "...requires frequent RP overflights to obtain updated transfer functions."

**L217: remove "the" before Appendix A;**
"The" has been removed.

**L224: "valueS";**
The typo has been corrected.

**L270-271: "ARE close to zero"**
The sentence has been corrected.

**Figure 5: Why is the symbol of NDII missing in the upper right panel?**
The second Reviewer had a similar comment in this regard. Figure 5 has now been revised with the markers in the correct position and a completed legend, and the intensity of the color scale has been adjusted to enhance the legibility of the markers.

**L281, 324 and other lines: I would refrain of using the term "exemplary"; it echoes a bit as "exemplary behavior or punishment"; Perhaps use "illustrates"?**
Based on the Reviewer's suggestion, all instances of "exemplary" have been rephrased.

**L345: Should be "The effect of changes in fdir";**
The text has been modified as suggested.

**L378-379: "… was between 0 and 40," and ".. ranged …"; Remove "were covered":**
The text has been modified as suggested.

**L406: remove comma after NDWI**
As suggested, the comma was removed.

**L407: twice increase, increasing;**
The second instance of "twice" was removed.

**Reply to Reviewer #3**
(Referee comment on "Biases in estimated vegetation indices from observations under cloudy conditions" by K. Wolf et al. (egusphere-2025-2082),[https://doi.org/10.5194/egusphere-2025-2082-RC2](https://doi.org/10.5194/egusphere-2025-2082-RC2), 2025)

Below, we respond to the Reviewer's comments. For clarity, the Reviewer's comments are in **bold** and the changes to the manuscript are *in italics*. Please note that some additional changes have also been made to improve the writing and style.
* * *
It is unfortunate that the methodology employed failed to convince the Reviewer. Nevertheless, we have incorporated all of the Reviewer's comments and suggestions into the manuscript. We hope these changes improved the manuscript in general and, in particular, addressed the Reviewer's concerns.

**GENERAL COMMENTS**
**This study makes use of a coupled atmospheric-vegetation radiative transfer model to simulate the effects of (homogeneous) clouds on three vegetation índices computed from drone imagery. To do so, the authors simulate a single canopy scenario with the optical radiative transfer model of SCOPE, and different illumination conditions (based on different combinations of sun zenith angle and cloud optical thickness). They present how the reflectance factors and the derived spectral índices are affected by these conditions and by changes in illumination between the measurement of reference panels and targets.**

**Despite the potential complexity of coupling two RTMs, which is part of another work under review, I fear I do not see any scientific added value in this work. I have the impresión that the results presented can be directly inferred from already known theory, and that the problem stated for the "drone" world has been dealt with for a long time in the area of field spectroradiometry. In fact, many of the statements referenced as findings from a former manuscript by the author (Wolf et al., 2024) are well-established in atmospheric radiative transfer and BRDF theory (e.g., "Therefore, $\rho(\lambda)$ is also sensitive to $\theta$ (see Wolf et al., 2024), Appendix D)" is already known in BRDF theory; also, the comments on the dependence of albedo on the fraction of direct/diffuse incoming irradiance are described, for example, by Pinty et al., 2005). Any novelty brought by this manuscript does not stand out from older theoretical knowledge, as presented.**

**At the same time, the range of simulated scenarios is so narrow (i.e., only one canopy) that the results presented cannot be generalized or used to provide any recommendations. In fact, the results could be different for canopies or surfaces of varying properties (e.g., see the effect of a bowl- or bell-shaped BRDF in Pinty et al., 2005). Since no generalizable knowledge can be obtained, the "cautions" arising from this work offer no added value to the knowledge already generated by the proximal sensing community. Additionally, the authors do not provide a drone-specific correction method, which, on the other hand, would already be known from field spectroscopy: continuous monitoring of irradiance and BRDF characterization, or just avoiding unstable conditions. For some of the scenarios described, proximal sensing would just discard the measurements due to changes in illumination. Maybe drone operators should**

**do the same. Moreover, drone-specific issues, such as the distance between the target and the sensor, where this modeling could offer something new (although this is also a long-known issue for airborne imagers), are often overlooked.**

The Reviewer argues that the effects of clouds on the estimated vegetation indices (VIs) discussed in the paper can be omitted by selecting cloud-free conditions or conditions with stable cloudiness. However, this requires continuous monitoring of the downward irradiance, particularly its individual components of direct and diffuse irradiance, since the direct-to-diffuse-ratio controls how radiation is reflected.

It also requires defining "stable" conditions. Jurado et al (1995), Perez et al. (2011), Calif et al. (2013), and Haaren et al. (2014) have shown that downward irradiance can vary within time scales from 300 s to as little as 1 s. Therefore, the definition of "stable" conditions is determined by the time intervals between reflectance panel (RP) overflights. It is further defined by the accepted uncertainty range of estimated VIs.

To our knowledge, the model coupling described by Wolf et al. (2025a) and the analysis presented in this manuscript are valuable tools for providing an initial estimate of the uncertainties induced by changes in cloud conditions.

Ultimately, external constraints, such as the availability of drones, limited time periods - dedicated campaign periods, or continuous measurements, may not allow to wait for cloud-free or "stable" conditions.

**While the authors acknowledge the limitations of the simulations, the justification provided - "Nevertheless, we argue that the presented study can be used as a first approximation for the transition between cloud-free and cloudy regions"- is insufficient a scientific publication; particularly when no new knowledge is generated, and the claimed sunny to cloudy transition (characterized by heterogeneous cloud cover as these develop or arrive from the horizon) is not represented by the simulation of homogeneous clouds they perform.**

We do admit that it is very unlikely that the cloud optical thickness suddenly increases from 0 to 40 or vice versa between RP overflights. In the previous version of the manuscript in Subsections "Effect of cloud changes on the normalized differential vegetation index (NDVI)" and "Effect of cloud changes on the normalized differential water index (NDWI)", we had selected these values for maximum contrast in cloud conditions, i.e., to provide the maximum envelope that would be expected from changes in cloud optical thickness.

To broaden the discussion about the variations in $\tau$, we added paragraphs to the subsections in "Effect of clouds and cloud changes on two-band vegetation indices".

Subsubsection "Effect of cloud changes on the normalized differential vegetation index (NDVI)" now includes:

*"The aforementioned extreme changes in $\tau$ from 0 to 40, and vice versa, between RP overflights are unlikely in the case of stratiform clouds. These extremes of $\tau$ have been selected to estimate the maximum envelope for biases in NDVI. Biases in NDVI, retrieved under more homogeneous conditions, will be smaller. However, even seemingly stratiform clouds vary in $\tau$ and therefore bias the retrieved values of NDVI. These variations have the greatest impact, when the RP overflight is performed under cloud-free conditions ($\tau_{cal} = 0$) or under conditions with small values of $\tau_{cal}$, since the slope of the curves is greatest in these situations (see, for example, the blue lines in Fig. 6a–c).".*

Subsubsection "Effect of cloud changes on the normalized differential water index (NDWI)" now includes:

*"As for the NDVI, the transitions in $\tau$ from 0 to 40, or vice versa, are extreme cases. For practical applications, variations in $\tau_{mea}$ for a given $\tau_{cal}$ are more relevant. For the $NDWI_{1640}$, the slopes in Fig. 6d–f are shifted in absolute terms but are constant for the same values of $\theta$. Therefore, variations in $\tau_{mea}$ between RP overflights will cause the same bias in $NDWI_{1640}$, independent of $\tau_{cal}$."*

We also invite the Reviewer to follow the changes in the track-changes file.

**I have the feeling that the authors have invested time in building a tool (Wolf et al., 2024), but they have not yet found a way to make a significant contribution with it, at least in this case. I recommend stepping back, reviewing former works on this topic, and identifying a problem that this tool can genuinely contribute to solving or alleviating.**

We thank the Reviewer for acknowledging the potential difficulties of combining two radiative transfer models, as described in Wolf et al. (2025a, 'Impact of stratiform liquid water clouds on vegetation albedo quantified by coupling an atmosphere and a vegetation radiative transfer model'). Based on the Reviewer's feedback, we understand the concerns regarding the scientific significance of the submitted  manuscript.

The Reviewer refers to the paper by Pinty et al. (2005), in which the influence of the atmosphere on the downward irradiance and its consequences for the bihemispherical reflectance (BHR) are discussed. There, the authors derive a mathematical framework to estimate the surface-reflected radiation and surface albedo over an anisotropic surface considering anisotropic atmospheric conditions. They discuss how a bowl- or bell-shaped bidirectional reflectance factor (BRF) modifies the surface albedo in combination with different aerosol loadings and solar zenith angles. Findings from Wolf et al. (2025a) and from the present manuscript are in agreement with Pinty et al. (2005), however, the latter one does not explicitly address the contribution of clouds, although clouds are briefly mentioned. The analysis presented by Pinty et al. (2005) is also limited to aerosol loadings up to $\tau=1$. Furthermore, the effects described in Pinty et al. (2005) are assumed to be monotonic over the selected wavelength ranges, missing the important wavelength dependent effects of scattering and absorption in the atmosphere, particularly the spectral effects from clouds. We argue that the present manuscript contributes to the scientific discussion about potential error sources in vegetation remote sensing and estimated VIs by systematically quantifying the deviations in estimated VIs under constant and changing cloudy conditions. Additionally, the paper can be used to estimate when certain VIs are insensitive to cloud influences.
The provided reference to Pinty et al. (2005) is valuable and we cite it in the present manuscript, for example in the introduction and in the appendix section. In addition, we added a sentence in Section "Influence of clouds on the spectral reflectance":
*"Pinty et al. (2005) have shown the influence of molecules and aerosols on direct and diffuse $F^{\downarrow}(\lambda)$, the effects on $F^{\uparrow}(\lambda)$, and the resulting albedo effects over vegetated areas based on analytical equations. Wolf et al. (2025a) used coupled atmosphere–vegetation radiative transfer models to investigate these effects, explicitly considering clouds and spectral depending scattering and absorption in the atmosphere.".*

The limitations of one-dimensional radiative transfer simulations in the atmosphere and vegetation are well-known. However, one-dimensional simulations provide an approximation without the requirement for complex, computationally expensive three dimensional simulations. This allows to use the combined model to easily extend the presented results by simulations of different vegetation types and other atmospheric settings. Furthermore, there is no model combination that holistically treats three-dimensional radiative transfer across atmosphere and vegetation in a way our coupled model is capable. The limitations of the one-dimensional simulations are already outlined in Wolf et al. (2025), as well as in the summary and conclusion section of the presented manuscript.

We agree with the Reviewer that some scenarios can be avoided by proper planning and calibration of drone observations. We think, that our simulations and the quantification of potential errors will help to make decisions about whether observations can be made in a certain cloud scenarios. Additionally, the simulations can provide a tool for correcting biased observations. However, we do not promote a correction method because doing so would be outside of the scope of the present study, since it would require observational validation of the correction.

Summarizing our reply, we see a contribution of our study to the scientific discussion by providing a quantification of potential errors in estimated VIs under cloudy conditions and changes in cloud condition. To enhance the relevance of the results, the simulations and evaluation was extend for results for the erectophile and the planophile LAD.

The following sentence was added in Section "Vegetation radiative transfer model SCOPE2.0":
*"The simulations were primarily considered for the general case of a spherical LAD, where all leaf angles have a similar probability (Goel et al., 1988). Further simulations have been performed to also include the erectophile and the planophile LADs."*

Additionally, a new section was added to the Appendix titled "Effect of cloud changes on the NDVI, $NDWI_{1640}$, and EVI for erectophile and planophile LADs". This section briefly discusses the effects of erectophile and the planophile LAD on biases in estimated NDVI, $NDWI_{1640}$, and EVI. Since a new section has been added, we ask the Reviewer to have a look through the track changes file.

Furthermore, the following paragraph was added to the "summary and conclusion" section:
*"The impact for other LADs, such as the erectophile and planophile, has been investigated. The erectophile LAD is more susceptible to biases in VIs under constant cloud conditions than the spherical LAD, while the bias in VIs with underlying planophile LAD is almost unaffected. Although the absolute value of the VI biases are determined by the LAD, the relative change due to a transition in τ between RP measurements was found to be constant among the three LADs."*

**References.**
**\* Pinty, B., Lattanzio, A., Martonchik, J. V., Verstraete, M. M., Gobron, N., Taberner, M., Widlowski, J.-L., Dickinson, R. E., and Govaerts, Y.: Coupling Diffuse Sky Radiation and Surface Albedo, Journal of the Atmospheric Sciences, 62, 2580-2591, https://doi.org/10.1175/JAS3479.1, 2005.**

**\* Wolf, K., Jäkel, E., Ehrlich, A., Schäfer, M., Feilhauer, H., Huth, A., Weigelt, A., and Wendisch, M.: Impact of clouds on vegetation albedo quantified by coupling an atmosphere and a vegetation radiative transfer model, EGUsphere, 2024, 1–30, https://doi.org/10.5194/egusphere-2024-3614, 2024.**

**\* Schaepman-Strub, G., Schaepman, M. E., Painter, T. H., Dangel, S., and Martonchik, J. V.: Reflectance quantities in optical remote sensing—definitions and case studies, Remote Sensing of Environment, 103, 27-42, https://doi.org/10.1016/j.rse.2006.03.002, 2006.**

**SPECIFIC COMMENTS**
**Reflectivity: The term reflectivity is misused (it is rather an intrinsic property of the matter, whereas the reflectance and related terms are sample-specific), and for sure, SCOPE does not simulate reflectivity. The right term to be used in this work is "reflectance factor" (see Schaepman-Strub et al., 2006 for a definition).**

We agree with the Reviewer that the term 'reflectivity' was used incorrectly in the paper. Where appropriate, we have replaced the word 'reflectivity' with 'reflectance'. However, 'reflectance factor', as suggested by the Reviewer, is not always the most appropriate term.

We calculated the reflectance using the radiance output from SCOPE and the downward irradiance from libRadtran.

**Equation 1: The term "sr" is not defined, "τ" is not defined in the subsequent paragraphs. Overall, make sure all terms are defined the first time they are introduced in the text.**

"sr" represents the unit steradian. We added an explanation after Eq.1 as following:
*"... sr the unit steradian, ..."*
The cloud optical thickness τ is already defined in lines 45-46 (submitted version for review) or lines 39-40 (in the updated manuscript).

**Lines 39-40: What reference supports this statement? Reflectance factors can be larger than one, but reflectance (the ratio of the total reflected to total incoming radiance) cannot (see again Schaepman-Strub et al., 2006). If so, are "broken clouds" what make reflectance factors go beyond one? If that has been proven, it must be referenced (and also other causes).**
According to the principle of conservation of energy, reflectance is confined to the interval [0, 1] (Schaepmann-Strub et al., 2006). As already mentioned in the manuscript, fields of broken clouds can cause the reflectance to exceed this interval; see, for example, Marshak et al. (2008). Only in heterogeneous cloud cases, where three-dimensional scattering effects occur, are reflectance values greater than one possible. In neighboring regions, reflectance is reduced. On average, this results in energy conservation.
To further clarify this, we added the reference of Marhsak et al. (2008). We further added the following sentences to the manuscript:
*"Only in heterogeneous cloud cases, where three-dimensional scattering effects occur, are reflectance values greater than one observable. In neighboring regions, reflectance is reduced. On average, this results in energy conservation."*

**Lines 41: Remote sensing does not retrieve vegetation índices. It retrieves reflectance factors, from which vegetation índices can be computed.**
To address the Reviewer's concerns, we replaced the word "retrieve" with "estimate". The sentence now reads:
*"Remote sensing techniques to estimate VIs from satellite measurements require cloud-free conditions."*

**Equation 4: Misses the scaling factor of the integration time or gains set for the sensor and ignores the dark current that must be subtracted from the recorded digital signal.**
We agree with the Reviewer that multiple factors and corrections are necessary for converting the recorded raw signal (digital counts) to radiance or reflectance. However, the specific corrections and factors applied depend on the system used. Here, we have provided a general equation that does not consider system-specific factors. Therefore, we refrain from including additional factors, such as those mentioned by the Reviewer, and to keep the equation as general as possible.

**Lines 69-74: This explanation is nuclear. There are several ways to prepare for a drone flight. Sometimes, the operator holds the camera above a reference panel (by the way, usually grey, and the darker the less Lambertian, assuming in the best case situation Spectralon® is used) to optimize the integration time and get a cross-calibration between the up-welling and the down-welling channels of the sensing sistema if the second exists, which is not always the case. The alternative is to place large reference panels in the study area and fly above them in one of the overpasses, using them as a reference. It is not clear what procedure the authors describe.**
While we agree that, in some instances, the downward radiation is also measured, this is only true in a few cases.

As we understand it, we are describing the same process that the Reviewer refers to as the 'alternative' method. In this method, reflectance panels are periodically overflown and used to calibrate the radiance sensor. According to the existing literature, this is an often used approach for determining surface / vegetation reflectance and subsequently estimating VIs. A schematic figure of the measurement strategy that we are addressing is provided in Fig.1 of the submitted and the revised versions of the manuscript.

The presented manuscript addresses that, in many cases of field observations, the downward irradiance is not directly measured. Instead, reflectance is indirectly inferred by using RP as a reference. Since no information about the downward irradiance is available, changes in the downward irradiance cause distortions in the spectral reflectance and in terms of absolute values. The manuscript's main purpose is to systematically quantify these effects and the consequences on estimated VIs.

**Lines 153-154: The fact that libRadtran simulates down-welling irradiance at 40 m above ground means that no differences between the irradiance level at the drone and at the target height are considered. This should be clarified, justified, and the potential impact on the results discussed.**

The difference in downward irradiance across the solar wavelength range (220–2500 nm) between the top of the atmosphere and the ground is primarily controlled by molecular absorption (mainly water vapor and ozone), scattering by aerosol particles and gas molecules, and clouds.

The impact of these factors on the downward irradiance in the lower 100 meters of the atmospheric column, which is relevant for the type of drone measurements discussed in the paper, is marginal compared to the influence of the remaining atmospheric column, extending from 100 to about 20 000 m.

Four simulations of spectral downward irradiance were performed representing flight altitudes of 0 m, 10 m, 40 m, and 100 m.

[Figure]

The left panel shows the spectral downward irradiance at each of the four levels. The resulting downward irradiance values are nearly identical, resulting in overlapping lines. The right panel shows the relative difference in downward irradiance at 0 m, 10 m, and 100 m altitude compared to the simulations at 40 m altitude. Regardless of the altitude or the wavelength, the relative differences are below 1%, except at the edges of the absorption bands. The relative differences are also in the typical range of measurement uncertainties.

The large relative errors result from the generally small absolute values of the downward irradiance, whereby even small absolute differences in the downward irradiance result in large relative differences. Values with low downward irradiance in the strong water vapor absorption bands have been partly masked out in the plots above, which is acceptable since these edges were also excluded in the analysis of the present manuscript. From the simulations it is concluded that there is no significant influence of the measurement altitude on the spectral downward irradiance.

**Lines 212-213: "While direct radiation is reflected into a narrow solid angle, diffuse radiation is scattered over a wider solid angle (Schaepman-Strub et al., 2006)." This statement is false. Direct radiation is reflected in all directions (e.g., a Lambertian diffuser).**
We partly agree with the Reviewer and admit that the sentence was imprecisely phrased. Therefore, we rephrased the sentence as follows:
*"In the case of non-isotropic surfaces, direct radiation is scattered in all directions with a significant portion of the radiation being scattered in a specific solid angle. Diffuse radiation is scattered almost equally over the entire hemisphere."*

**A different thing is that a Bi-directional reflectance factor is defined as the ratio between the incoming and outgoing radiance in infinitesimal solid angles.**
We assume that the Reviewer refers here to the discrepancy between the definition of bidirectional reflectance distribution function (BRDF) and real measurements that cannot provide radiance in infinitesimal solid angles. We refer to this practical issue in Appendix Section B of the manuscript, where it is stated that in practical applications only the hemispherical-directional reflectance factor (HDRF) can be measured.

**References**
Asner, G. P. et al., Biophysical and Biochemical Sources of Variability in Canopy Reflectance, 1998, *Remote Sens. Environ.* , Vol. 64, No. 3, p. 234-253

Marshak, A. et al., A simple model for the cloud adjacency effect and the apparent bluing of aerosols near clouds, 2008, *J. Geophys. Res. Atmos.* , Vol. 113, No. D14